# 3DPAN-CIL: A PROTOTYPE ASSISTED NETWORK OF CLASS-INCREMENTAL LEARNING FOR 3D POINT CLOUDS

## ABSTRACT

In response to the continuous influx of 3D point clouds encountered in practical training scenarios, we propose a novel incremental learning and classification approach designated as 3DPAN-CIL, specifically tailored for 3D point cloud data. This method initially establishes the 3D category prototype that encapsulates the feature embedding of point clouds within a latent space. Then, we wisely construct an optimal transport strategy on this prototype space for the migration of 3D category prototypes. This alignment ensures that the distribution of new category prototypes adheres as closely as possible to the relative spatial distribution of old category prototypes, significantly reducing the catastrophic forgetting in the training model. Additionally, to tackle the challenge of imbalanced old and new samples, we introduce a prior-guided knowledge distillation strategy aimed at addressing the model's preference for new knowledge. We conduct a series of experimental evaluations on both synthetic datasets and real scanning datasets, demonstrating that our method surpasses existing state-of-the-art approaches in terms of average accuracy and average forgetting rate. Notably, in the context of average scene partitioning, our method achieves improvements of 4.5% in average accuracy and 1.47% in average forgetting rate compared to other top-performing methods. The model and code are available at: `https://github.com/FlRiver/3DPAN-CIL`.

## 1 INTRODUCTION

Nowadays, notable advancements have been made in areas such as autonomous driving, scene analysis, and robotics, positioning 3D object classification with continual learning as one of paramount tasks within the realm of 3D visual technology (Chen et al., 2023). The rapid development of 3D acquisition devices, such as LiDAR, has facilitated the proliferation of point cloud data, which is distinguished by its straightforward representation and accessibility. Consequently, extensive research has been undertaken on classification utilizing incremental data derived from raw point clouds, significantly advancing data-driven deep learning methods (Liang et al., 2024a).

When dealing with large-scale 3D point cloud data, the transfer learning approach has been proposed to mitigate challenges such as extended training durations and the scarcity of new data, demonstrating its efficacy in tasks with significant relevance. Furthermore, the application of pre-trained point cloud models effectively addresses a majority of classification problems. Nevertheless, when confronted with incrementally emerging data streams, existing models often suffer from catastrophic forgetting (Yu et al., 2022; Pang et al., 2022), characterized by a rapid adaptation to new classes at the cost of previously acquired knowledge. Conversely, an excessive focus on alleviating catastrophic forgetting may impede the adequate assimilation of new class data, diminishing the classification performance for new categories.

Recent advancements in continual learning for images have been notable (Zhu et al., 2022; Yan et al., 2021; Pham et al., 2021). However, models designed for 3D point clouds have not demonstrated satisfactory performance in class-incremental learning, primarily due to three key factors. First, during the training process, models tend to overfit the distribution of the current data, which result in the model forgetting the distribution of previous data. While distinguishing between old and new tasks through the training network extension appears to be a viable strategy, the model's size tends to increase with the number of tasks, which poses challenges for the deployment and practical application of learning models. Second, although the distribution of old category prototypes is maintained, there is a significant disparity of prototypes between new and old categories in the

training model, which adversely impacts the model's generalization capabilities concerning new categories. Furthermore, a considerable amount of current research employs a replay-based approach to address the issue of forgetting in class-incremental learning. This method involves allocating a limited memory space to retain data from old categories. However, it could result in an imbalance between the quantities of new and old sample data, leading the model's classifier to favor the new category.

In this study, we propose a novel incremental learning and classification network (3DPAN-CIL), aimed at mitigating the significant performance degradation when confronted with continuous streams of 3D point clouds. The primary contributions of this research are: (1) We propose an effective class prototype space construction in 3D class-incremental learning. It applies point cloud position and normal with Transformer based module to solve the unorderness and irregularity of 3D models with noise, partial missing and geometric occlusion (specific to 3D models). (2) We introduce the batch-wise optimal transport (OT) on this prototype space and successfully solve catastrophic forgetting in 3D incremental learning by utilizing old class prototype space as a directional guide and adjusting the class prototype migration in new prototype space. (3) We derive prior guided knowledge and apply dynamic weighting to address the data bias inherent in the training model. Subsequently, by distilling knowledge from both balanced labels and soft labels, we enhance the new model's ability to assimilate established knowledge from the previous model.

## 2 RELATED WORK

**2.1 3D Point cloud classification.** The evolution of 3D point cloud acquisition technologies has resulted in the creation of various models (e.g., PointNet (Qi et al., 2017a), PointNet++ (Qi et al., 2017b)) that directly analyze raw point cloud data. Models such as PointCNN (Li et al., 2018), PointConv (Wu et al., 2019a), and DGCNN (Wang et al., 2019) have followingly emerged, with other methods increasingly emphasizing enhancements through attention mechanism, Transformer and Mamba (Liang et al., 2024a). For instance, PCT (Guo et al., 2021) encodes point cloud features into higher-dimensional spaces and employs multi-layer self-attention and biased attention modules to capture multi-scale semantic similarities for classification. Point-BERT (Yu et al., 2022) introduces a masked point modeling to train point cloud Transformers, while utilizing an additional dVAE to generate discrete token representations of point clouds. Point-MAE (Pang et al., 2022) processes point clouds within the masked autoencoder, relying exclusively on Transformers without supplementary frameworks. PointGPT (Chen et al., 2023) employs a hierarchical Transformer architecture that integrates dynamic graph neural networks to enhance feature representations for downstream analysis. However, these methods are not readily applicable to class-incremental learning tasks involving 3D point clouds, as they are susceptible to catastrophic forgetting when faced with the continuous introduction of new 3D point clouds.

**2.2 Class-incremental learning.** Class-incremental learning has recently attracted considerable scholarly interest. To tackle the issues of plasticity and catastrophic forgetting in models, four primary strategies have been proposed: regularization-based methods, knowledge distillation-based methods, network architecture-based methods, and replay-based methods. The predominant approaches tend to emphasize knowledge distillation and replay strategies. Kirkpatrick et al. (2017) introduced an elastic weight consolidation (EWC) model, which can alleviate catastrophic forgetting by constraining critical parameters. Li & Hoiem (2017) developed a learning without forgetting (LwF) model that incorporates the knowledge distillation loss as a fundamental element in numerous continual learning frameworks. Rebuffi et al. (2017) introduced a replay-based incremental learning method iCaRL, which selects a certain number of samples, and then combines the preserved samples with new data for joint training. Zhu et al. (2021) proposed a prototypical augmented self-supervised (PASS) method, which maintains class prototypes for each category and enhances the training dataset by incorporating Gaussian noise. Furthermore, it employs a self-supervised learning strategy to mitigate the risk of the feature extractor overfitting to new classes.

**2.3 Incremental learning of 3D point clouds.** Current investigations in class-incremental learning primarily concentrate on image domains, presenting significant challenges when attempting to 3D point clouds. Compared to 2D images, incremental learning for 3D point clouds meets following issues: (1) Unorderness and irregularity in 3D point clouds with noise, (2) Geometric preservation in 3D data with incompleteness and occlusion (Qi et al., 2025; Xiang et al., 2025). How to preserve geometric feature and relationship is the main challenge in the incremental learning of 3D point clouds. Liu et al. (2021) introduced an L3DOC model, which disaggregates feature extraction modules on a layer-wise basis to effectively capture shared point knowledge, thereby addressing the issue of catastrophic forgetting. Chowdhury et al. (2021) developed an LwF-3D model which utilizes class semantic embeddings from previous models to generate soft labels that guide updates in new models. Dong et al. (2021) proposed an I3DOL model, which constructs geometry-aware centroids

through attention mechanisms to selectively identify local regions and maintains an optimal exemplar set to mitigate forgetting. Zamorski et al. (2023) introduced a RCR model which employs the compressed point cloud sampling for replay and integrates the reconstruction loss as a regularization term. Xu et al. (2025) provided an enhanced 3D few-shot class-incremental learning by applying the vision-language models. But it is related to the pre-training which greatly increases the model complexity. In contrast to these methods, our 3DPAN-CIL enhances the learning of new categories by optimizing model updates through alignment in the latent prototype space, which can significantly reduce catastrophic forgetting while preserving robust plasticity for the incorporation of new classes.

## 3 METHODOLOGY

**3.1 Problem definition.** Given a specific task sequence $T \in \{1, 2, \cdots, t\}$, the training samples are denoted as $D_T \in \{d_1, d_2, \cdots, d_t\}$. The samples and corresponding labels utilized for model training are restricted to the currently accessible data, denoted as $d_t \in \{x_t^i, y_t^i\}$. In this context, $d_t$ represents the training dataset at the $t$-th stage, $x_t^i$ signifies the $i$-th data within the $t$-th stage of the task, $y_t^i$ indicates the $i$-th label associated with the $t$-th stage of the task, and $x^i \in X, y^i \in Y$. In contrast to conventional training methods that permit access to the entirety of the training dataset, class-incremental learning is constrained to the utilization of a minimal set of training samples pertinent to the current phase. Furthermore, the memory capacity $M$ allocated for samples from old categories adheres to $|M| \ll N_D$, where $N_D$ denotes the total number of examples within the training dataset.

**3.2 Overview of overall framework.** The overall framework of our 3DPAN-CIL is illustrated in Figure 1. In the context of the incremental $t$-task phase, the model is restricted from utilizing the data associated with the previous categories ($D_0, \cdots, D_{t-1}$), which is maintained in a consistent manner with the feature extraction component of Point-BERT (Yu et al., 2022) using positional encoding. In addition, it makes full use of mask modeling, local information and attention mechanism to improve the feature extraction ability of 3D point clouds. Specifically, all samples are firstly transformed into feature embeddings through Mini-Point (Qi et al., 2017b), resulting in a dimensionality with $\mathbf{Z}_{in} \in \mathbb{R}^{g \times k}$, where $g$ is the number of sub-point clouds obtained by applying the farthest point sampling (FPS) method to a single point cloud model, $k$ represents the dimensional increase of point cloud data from three-dimensional representation to a higher $k$-dimensional space, $\mathbf{Z}_{in}$ is the feature obtained from Mini-point. Then we develop a dual-branch feature extractor comprising both an old model and a new model. The new model is initialized based on the parameters of the old model, which remains static during the feature extraction process. The output feature $\mathbf{Z}_{out}$ is achieved through a feature extractor $f$ that incorporates standard Transformer blocks (Vaswani et al., 2017) and combined poolings. It can effectively solve the unorderness and irregularity of 3D models with noise, partial missing and occlusion. Initially, we establish category prototypes to encapsulate the representative features of each class of point clouds. An optimal transport loss is then calculated to facilitate the migration within the category prototype space, which effectively regulates the updates to the new model. Given the challenge posed by data imbalance between old and new samples in practical scenarios, we apply weights to the predicted labels informed by prior knowledge regarding the sample sizes of both old and new stages. Furthermore, we implement a distillation strategy to facilitate the transfer of knowledge from the old model to the new model.

**3.3 Spatial migration of 3D category prototypes.** Presently, numerous methods in class-incremental learning aim to minimize the disparity between old and new models by employing KL divergence across each feature extraction layer in both models. However, this sample-level distillation is hindered by the off-centre distillation issue (Tang et al., 2020). In response, we propose a category space-level distillation approach to mitigate the bias problem, emphasizing that the overall distribution of the new task's category space should align with that of the old task. Furthermore, in high-dimensional spaces, the direction of migration can vary among different category prototypes throughout the migration process. To ensure that each category prototype attains its optimal position, we implement an optimal transport strategy that progressively aligns each category prototype with the category distribution of the old task, illustrated in Figure 1(a).

**3.3.1 3D category prototype of point clouds.** We define the category prototype set of 3D point clouds as $\mathbf{P} = \{\mathbf{p}_1, \mathbf{p}_2, \cdots, \mathbf{p}_N\}$, where each category prototype is designed to represent each type within 3D point clouds, based on a specified set of high-dimensional features $\mathbf{P} \in \mathbb{R}^{N \times k}$. In the context of incremental tasks, we posit that the category prototype associated with the old task, $t - 1$, is represented as $\mathbf{P}_{t-1} \in \mathbb{R}^{n_{t-1} \times k}$, while the category prototype for the current task, $t$, is denoted as $\mathbf{P}_t \in \mathbb{R}^{n_t \times k}$. The category increment for the new task is identified as $n_t - n_{t-1}$, which encompasses both the old category prototype $\mathbf{P}_t^{\text{old}}$ and the new category prototype $\mathbf{P}_t^{\text{new}}$, collectively referred to

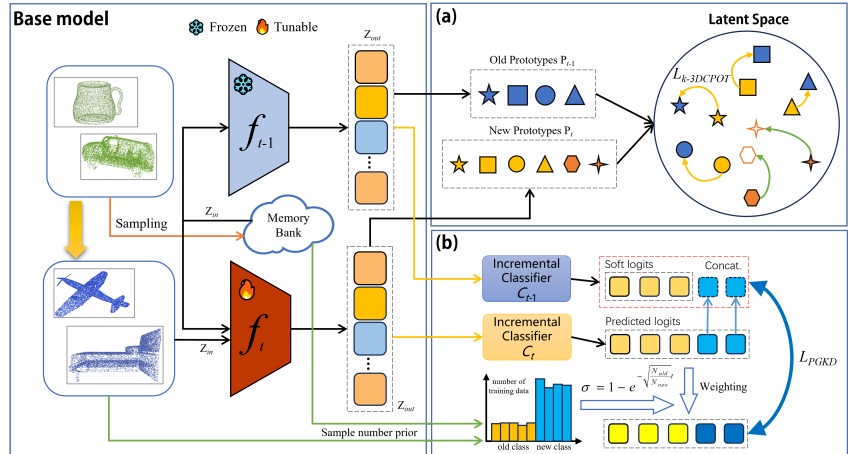

Figure 1: Overall framework of 3DPAN-CIL. The base model is the main structure of the framework, where $f_{t-1}$ and $f_t$ are feature extraction modules in previous and current stages respectively. Among them, $f_{t-1}$ is frozen and used to guide the training of $f_t$. Memory bank stores samples from time $t-1$ and combines the sample size at time $t$ to jointly calculate prior knowledge. (a) represents the OT process for class prototypes, where the update direction is guided by the OT loss $\mathcal{L}_{k\text{-3DCPOT}}$. (b) represents the knowledge distillation process guided by prior knowledge, where prior knowledge is used for weighting predicted logits to address the issue of imbalanced training data, and $\mathcal{L}_{\text{PGKD}}$ loss is used to guide the model training.

as $\mathbf{P}_t$. Subsequently, we employ the optimal transport method to assess $\mathbf{P}_t^{\text{old}}$ and to ascertain the variance between $\mathbf{P}_t^{\text{old}}$ and $\mathbf{P}_{t-1}$. Namely, we calculate the distance between $\mathbf{P}_t^{\text{old}}$ and $\mathbf{P}_{t-1}$ at the stage t, simultaneously applying OT in high-dimensional space to narrow the difference between $\mathbf{P}_t^{\text{old}}$ and $\mathbf{P}_{t-1}$. Due to $\mathbf{P}_{t-1}$ being the prototype space from the previous stage and being frozen (not changing in high-dimensional space), $\mathbf{P}_t^{\text{old}}$ is calculated in real-time and migrate as the model is trained. Therefore, we use the OT method to guide $\mathbf{P}_t^{\text{old}}$'s migration, preventing the model from forgetting old categories. For $\mathbf{P}_t^{\text{new}}$, it is imperative to maintain the relative spatial relationship with $\mathbf{P}_t^{\text{old}}$, which necessitates the adjustment of the appropriate position in accordance with $\mathbf{P}_t^{\text{old}}$.

**3.3.2 Optimal transport for 3D category prototypes of point clouds** The objective of this study is to incorporate the OT method into our model for mitigating catastrophic forgetting through the transfer of category prototypes. When the OT method is employed to quantify the disparity between distributions of 3D point clouds, the core task is to calculate the minimum total cost required to convert the difference between two distributions using Wasserstein distance. Cuturi (2013) proved that when two distributions are represented in a discrete form, the Sinkhorn distance can be used as a convenient computational form to obtain an approximate optimal solution. In fact, we can prove that in a more general case, when two distributions are represented in a continuous form, the Sinkhorn distance still satisfies the distance axiom condition for the approximate optimal solution of the OT problem. Please refer to Appendix A for the detailed proof.

For two point cloud distributions, denoted as $\mathbf{u}_p \in \mathbb{R}^m$ and $\mathbf{v}_p \in \mathbb{R}^m$, the difference between them is articulated through the Sinkhorn distance associated with $\mathbf{u}_p$ and $\mathbf{v}_p$, as follows

$$W_{\epsilon,p}^p(\mathbf{u}_p, \mathbf{v}_p) = \inf_{\pi \in \Pi_\epsilon(\mathbf{u}_p, \mathbf{v}_p)} \int_{\mathbb{R}^k \times \mathbb{R}^k} d(\mathbf{u}_p, \mathbf{v}_p) \, d\pi, \tag{1}$$

where $\pi$ denotes the joint distribution function corresponding to $\mathbf{u}_p$ and $\mathbf{v}_p$, $\Pi_\epsilon(\mathbf{u}_p, \mathbf{v}_p)$ represents the set of all joint distributions whose marginal distributions are $\mathbf{u}_p$ and $\mathbf{v}_p$ respectively, which satisfies the KL divergence constraint (Kim et al., 2021) as

$$\Pi_\epsilon(\mathbf{u}_p, \mathbf{v}_p) = \{\pi \mid \text{KL}(\pi \| \mathbf{u}_p \otimes \mathbf{v}_p) \leq \epsilon, \pi \in \Pi(\mathbf{u}_p, \mathbf{v}_p)\}, \tag{2}$$

$$\text{KL}(\pi \| \mathbf{u}_p \otimes \mathbf{v}_p) = \iint \pi \ln\left(\frac{\pi}{\mathbf{u}_p \mathbf{v}_p}\right) dx dy. \tag{3}$$

$d(\mathbf{u}_p, \mathbf{v}_p)$ is the transport cost function from $\mathbf{u}_p$ to $\mathbf{v}_p$, formulated as the Euclidean distance.

In the context of class-incremental learning, we conceptualize the distributions associated with two incremental stages, $t-1$ and $t$, as distributions $\mathbf{u}$ and $\mathbf{v}$ within two distinct latent spaces. Here, the distribution $\mathbf{u}$ at stage $t-1$ is designated as the shared category distribution, whereas the distribution at stage $t$ encompasses both the shared categories and newly introduced categories. Accordingly, we articulate the decomposition of the spatial distribution $\mathbf{v}$ at stage $t$ in a manner of $\mathbf{v} = \mathbf{v}_{\text{new}} \oplus \mathbf{v}_{\text{old}}$, where $\oplus$ signifies the dimensional splicing of categories, $\mathbf{v}_{\text{new}}$ indicates the distribution of the new categories at stage $t$, and $\mathbf{v}_{\text{old}}$ denotes the distribution of the old categories at time $t$, which is shared with $\mathbf{u}$. We adopt a replay based strategy in stage t, namely, retaining a small number of old samples for joint training. In the stage $t$, we can separate new and old category prototypes by prior knowledge in advance, and there is no the correspondence ambiguity of prototypes from stages in $t$-1 and $t$. The objective of our study is to minimize the discrepancy between $\mathbf{v}_{\text{old}}$ and $\mathbf{u}$, which can be conceptualized as an OT problem.

In the context of utilizing deep features $\mathbf{Z}_{t-1}$ and $\mathbf{Z}_t$ derived from the dual branch, we proceed to calculate their respective category prototypes, denoted as $\mathbf{P}_{t-1}$ and $\mathbf{P}_t$. Given that the new category samples of point clouds are employed during the new task phase and are absent in the previous task phase, we accordingly partition $\mathbf{P}_t$ into $\mathbf{P}_t^{\text{old}}$ and $\mathbf{P}_t^{\text{new}}$, which is represented as $\mathbf{P}_t = \mathbf{P}_t^{\text{old}} \oplus \mathbf{P}_t^{\text{new}}$, where the dimensions of $\mathbf{P}_t^{\text{old}}$ and $\mathbf{P}_{t-1}$ are equivalent to $\mathbb{R}^{n_{t-1} \times k}$, and the dimension of $\mathbf{P}_t^{\text{new}}$ is represented by $\mathbb{R}^{(n_t - n_{t-1}) \times k}$.

During the training process, due to the substantial size of the dataset and the constraints imposed by the memory capacity, we employ a batch processing approach. This entails loading only a subset of the training samples into memory for processing at any given time. Consequently, obtaining the point cloud distribution $\mathbf{P}_t$ at a specific time $t$ becomes challenging, necessitating the use of an accumulated approximation to address this issue. Initially, we define the category prototype of point clouds for the old task phase as follows

$$\mathbf{p}_{t-1}^i = \frac{1}{N_i} \sum_{j \in y_{\text{old}}^i} f_{t-1}(\mathbf{x}_j^i), \quad \mathbf{p}_{t-1}^i \in \mathbf{P}_{t-1}, \tag{4}$$

where $f_{t-1}$ represents the feature extractor utilized in the old model to obtain deep features, $N_i$ denotes the quantity of features associated with class $i$. Subsequently, the corresponding category prototype of each class, denoted as $\mathbf{p}_{t-1}^i$, is derived by averaging all these deep features. Following this, we introduce our OT method (3DCPOT), specifically designed for the category prototype of point clouds:

$$\text{3DCPOT}(\mathbf{P}_{t-1}, \mathbf{P}_i) \triangleq \text{OT}(\mathbf{P}_{t-1}, \mathbf{P}_t^{\text{old}}) = \min_{\pi \in \Pi_\epsilon(\mu, \nu)} \sum_{k=0}^{n_{t-1}} \sum_{l=0}^{n_{t-1}} \pi_{kl} d(\mathbf{p}_{t-1}^k, \mathbf{p}_t^l), \tag{5}$$

where $f_t$ serves as the feature extraction module. The cost function is formulated based on the distance cost metric $d(\mathbf{p}_{t-1}^k, \mathbf{p}_t^l)$, and a cost matrix is established utilizing the distance cost metric $d$, which is designed to quantify the expense associated with transporting each category in the spatial distribution of existing categories to corresponding categories in the new spatial distribution. In the training stage t, we follow the replay-based class incremental learning strategy, which always allows the model to store a small number of samples in the memory space for training, without the problem of no corresponding samples in the $t$-1 stage.

Despite the fact that our 3DCPOT is designed for category prototypes within two stages, we encounter a considerable limitation in memory capacity, which restricts our ability to process training samples in a single batch throughout the training phase. In response, we enhance the 3DCPOT by maintaining the category prototype of point clouds from the old stage in a fixed state. Specifically, we denote the current training sample size as $N$ and the batch size as $B$, leading to the number of iterations being represented as $K = N/B$. We define $\mathbf{Q}_i$ as the category prototype derived from the $i$-th batch of randomly selected sample data during the iteration, ensuring that there is no overlap between samples in each batch for the computation of $\mathbf{Q}$. Consequently, the optimal transport of the batch point cloud category prototype($k$-3DCPOT) can be expressed as

$$k\text{-3DCPOT}(\mathbf{P}_{t-1}, \mathbf{P}_i) \triangleq \frac{1}{K} \sum_{i=1}^{K} \text{OT}(\mathbf{P}_{t-1}, \mathbf{Q}_i), \tag{6}$$

In pursuit of minimizing the disparity between $\mathbf{P}_{t-1}$ and $\mathbf{P}_t^{\text{old}}$, we propose a 3D category prototype loss function ($\mathcal{L}_{k\text{-3DCPOT}}$) for training, which is articulated as

$$\mathcal{L}_{k\text{-3DCPOT}} = \min_F F_{k\text{-3DCPOT}}(\mathbf{P}_{t-1}, \mathbf{P}_t^{\text{old}}), \tag{7}$$

**3.3.3 Category prototype migration.** Due to the absence of corresponding prototypes for $\mathbf{P}_t^{\text{new}}$ in the latent space during the old task phase, and considering the spatial relationship between $\mathbf{P}_t^{\text{new}}$ and $\mathbf{P}_t^{\text{old}}$ throughout the training process, we initiate the migration of latent features for $\mathbf{P}_t^{\text{new}}$ associated with category prototypes. Specifically, we update the position of $\mathbf{P}_t^{\text{new}}$ to maintain the relative positional relationship between old and new categories in the new task phase. As we know, for $\mathbf{P}_t^{\text{old}}$, each old category prototype is assigned a distinct direction for updating. However, there is a lack of precise guidance for each new category prototype in $\mathbf{p}_t^{\text{new}}$. We can regard the old category prototype distribution $\mathbf{P}_t^{\text{old}}$ as a reference for the whole latent space distribution to describe the guidance migration, which can easily convert the migration of new category prototypes in $\mathbf{P}_t^{\text{new}}$ into the migrating process along with $\mathbf{P}_t^{\text{old}}$ in the latent space.

Given that category prototypes of 3D point clouds exist in a high-dimensional space, where each dimension corresponds to a distinct direction, we employ a distance metric that encompasses all dimensions of each prototype in $\mathbf{P}_t^{\text{old}}$. This feature-level metric computation in each dimension ensures that $\mathbf{P}_t^{\text{new}}$ more accurately approximates the optimal migration of $\mathbf{P}_t^{\text{old}}$. Specifically, we posit that $\mathbf{P}_{t-1}$ comprises $n_{t-1}$ category prototypes and $\mathbf{P}_t$ encompasses $n_t$ category prototypes, all sharing the same dimensionality $k$. Consequently, our transport cost function is articulated as

$$\mathbf{d} = \frac{1}{n_{t-1}} \mathbf{1}_{n_{t-1}} (\mathbf{P}_t^{\text{old}} - \mathbf{P}_{t-1}), \quad \mathbf{d} \in \mathbb{R}^k, \tag{8}$$

where $\mathbf{1}_{n_{t-1}}$ is an $n_{t-1}$-dimensional row vector consisting entirely of ones, and $\mathbf{d}$ is the comprehensive optimal transport metric vector that indicates the average update direction for all old category prototypes within dimension $k$. Ultimately, $\mathbf{d}$ is employed to update $\mathbf{P}_t^{\text{new}}$ as a migration reference. Specifically, for each prototype $j$ in $\mathbf{P}_t^{\text{new}}$, the update is conducted as $\tilde{\mathbf{P}}_{t,j}^{\text{new}} = \mathbf{P}_{t,j}^{\text{new}} + \beta \cdot \mathbf{d}$, where $\beta$ is the weight adjustment parameter utilized in the update direction. Upon the completion of the migration process, we combine $\tilde{\mathbf{P}}_t^{\text{new}}$ and $\mathbf{P}_t^{\text{old}}$ to form $\tilde{\mathbf{P}}_t$, which subsequently serves as a guiding framework for model training at the subsequent time $t+1$, denoted as $\tilde{\mathbf{P}}_t = \mathbf{P}_t^{\text{old}} \oplus \tilde{\mathbf{P}}_t^{\text{new}}$.

**3.4 Priori guided knowledge distillation.** In the context of incremental learning, the phase associated with new tasks lacks access to samples from previous categories and is restricted to utilizing a limited number of old samples retained in memory. This situation results in an imbalance between the quantities of samples from old and new stages, which is likely to lead to a classifier that prioritizes the acquisition of knowledge related to the new task and consequently results in the erosion of previously acquired knowledge pertaining to old categories. To mitigate this imbalance, we propose leveraging the quantities of old and new samples as a form of priori knowledge.

**3.4.1 Old and new sample size as a priori.** When quantifying the number of new samples ($N_{\text{new}}$) during the training phase and the number of old samples ($N_{\text{old}}$) stored in the computer's memory, we introduce $\lambda = \sqrt{N_{\text{old}}/N_{\text{new}}}$ to represent the sample size-based prior knowledge. Given that the memory capacity is constant and the representation of each category of old samples within this memory diminishes over time, we establish a concept of prior knowledge $\sigma = 1 - e^{-\lambda t}$, which serves to effectively mitigate the impact of temporal changes.

**3.4.2 Dynamic weighting.** For the classifier $C_t$ designed for the new task, given that $N_{\text{old}} \ll N_{\text{new}}$, the output logits are predominantly influenced by the training of new category samples. This situation leads to an issue of imbalanced data distribution throughout the model training process. We apply weighting to the predicted logits in order to recalibrate the model's emphasis on both old and new categories, as illustrated in Figure 1(b). We denote the output of $C_t$ as $S_{\text{new}} \in \mathbb{R}^{n_t}$ and the output of $C_{t-1}$ as $S_{\text{old}} \in \mathbb{R}^{n_{t-1}}$. Due to the differing dimensions of $S_{\text{new}}$ and $S_{\text{old}}$, $S_{\text{new}}$ is divided into two components as $S_{\text{new}} = S'_{\text{new}} \oplus S''_{\text{new}}$, where $S'_{\text{new}}$ and $S''_{\text{new}}$ represent the logit probabilities associated with old and new categories produced by $C_t$. Subsequently, we employ the sample size prior $\sigma$ for $S'_{\text{new}}$ and $S''_{\text{new}}$ to achieve a balanced emphasis within the model on both old and new categories, which is expressed as $\tilde{S}_{\text{new}} = \sigma S'_{\text{new}} \oplus (1 - \sigma) S''_{\text{new}}$.

**3.4.3 Priori guided knowledge distillation loss.** Like established methods (Wen et al., 2024; Kang et al., 2022; Shang et al., 2023), we incorporate a knowledge distillation loss (Hinton et al., 2015) to mitigate the problem of knowledge forgetting. Nonetheless, prior methods in knowledge distillation

have overlooked the issue of sample imbalance, opting instead to utilize KL divergence for soft and predictive labels. This oversight would lead to challenges such as overfitting and underfitting in training. We propose a priori guided knowledge distillation loss, which preserves the retention of old knowledge by formulating the predictive label $\tilde{S}_{\text{new}}$ with dynamical weighting as a form of prior knowledge. Note that the dimensionality of $S_{\text{old}}$ derived from the previous model is less than that of $\tilde{S}_{\text{new}}$, rendering direct application of knowledge distillation for loss calculation unfeasible. In a similar vein, we concatenate the components of $S''_{\text{new}}$ and $S_{\text{old}}$ to ensure the dimensional consistency $\tilde{S}_{\text{old}} = S_{\text{old}} \oplus S''_{\text{new}}$. Finally, we define a priori guided knowledge distillation loss as follows

$$\mathcal{L}_{\text{PGKD}} = \mathcal{L}_{\text{KL}}(\tilde{S}_{\text{new}}, \tilde{S}_{\text{old}}), \tag{9}$$

where the KL divergence is used to measure the gap between $\tilde{S}_{\text{new}}$ and $\tilde{S}_{\text{old}}$.

**3.5 Loss function and parameter setting.** In the initial phase of model training, we employ the cross-entropy loss ($\mathcal{L}_{\text{CE}}$) as the primary loss function within the foundational incremental stage of our model. In the subsequent training phase, we incorporate losses $\mathcal{L}_{k\text{-3DCPOT}}$ and $\mathcal{L}_{\text{PGKD}}$ alongside loss $\mathcal{L}_{\text{CE}}$ to facilitate the co-optimization of the model training as denoted by

$$\mathcal{L} = \mathcal{L}_{\text{CE}} + \alpha_1 \mathcal{L}_{k\text{-3DCPOT}} + \alpha_2 \mathcal{L}_{\text{PGKD}}, \tag{10}$$

where $\mathcal{L}_{\text{CE}}$ is the cross-entropy loss, $\mathcal{L}_{k\text{-3DCPOT}}$ is the optimal transport loss associated with the category prototype of point clouds as detailed in Eq. 7, and $\mathcal{L}_{\text{PGKD}}$ refers to the priori-guided knowledge distillation loss discussed in Eq. equation 9. $\alpha_i$ ($i = 1, 2$) is the hyperparameter corresponding to each loss, in which we set $\alpha_1 = \alpha_2 = 5$ across all experimental conditions. The main procedure of our 3DPAN-CIL is delineated in Appendix B. In addition, we provide the complexity analysis of our model, which is described in Appendix C.

The experiments are conducted on an NVIDIA RTX A6000 GPU server. In alignment with established point cloud classification methods (Yu et al., 2022; Pang et al., 2022; Liang et al., 2024b), the experiments utilize the original point cloud data, with the dataset being segmented according to the incremental learning task for class categories. The AdamW optimizer is employed to optimize the neural network, with a learning rate fixed at 0.001, and the CosLR learning rate scheduler is utilized to adjust the learning rate. A batch size of 256 is established, with training conducted over 100 iterations for each incremental stage and global random seed of 1229 is applied. The dataset loading is configured in accordance with the Point-BERT model (Yu et al., 2022), and the parameter tuning for $\alpha_i$ is elaborated in Appendix D.1.

## 4 EXPERIMENTAL RESULTS

**4.1 Datasets.** The experimental dataset utilized in this study comprises five commonly used datasets according to data scale, model type and application scenario: ModelNet (Wu et al., 2015), ShapeNet (Chang et al., 2015), ScanObjectNN (Uy et al., 2019), ScanNet (Dai et al., 2017) and CO3Dv2 (Reizenstein et al., 2021). Among these, ModelNet and ShapeNet are classified as synthetic datasets, whereas ScanObjectNN, ScanNet and CO3Dv2 are categorized as real scanning datasets. The detailed dataset description is provided in Appendix D.2. We establish a benchmark for dividing the dataset according to incremental categories. This process involves the implementation of two distinct experimental scenarios: the category averaging principle (e.g., ModelNet40-avg) and the over half partitioning principle (e.g., ModelNet40-half). The latter principle is further examined in the context of ablation experiments. Under the category averaging principle and over half principle, we perform five related tests respectively. The detailed dataset partitioning description with averaging principle is depicted in Appendix D.3.

**4.2 Comparison method.** We employ the method comparison to evaluate prevalent approaches of incremental learning within the context of 3D point clouds. Additionally, recognizing that certain research has concentrated on experimental frameworks involving 2D images, we also juxtapose our model against these methods, as long as we replace the features extracted from the image by 3D point cloud features as data input. A total of 11 advanced methods have been selected for this comparative analysis, including LwF-3D (Chowdhury et al., 2021), I3DOL (Dong et al., 2021), LwF (Li & Hoiem, 2017), iCARL (Rebuffi et al., 2017), EEIL (Castro et al., 2018), BiC (Wu et al., 2019b), WA (Zhao et al., 2020), GeoDL (Simon et al., 2021), CafeBoost (Qiu et al., 2023), EASE (Nishikawa et al., 2022) and DECO (Luo et al., 2024). Furthermore, we present two benchmark comparisons (Li & Hoiem, 2017): Joint-Training and Fine-Tuning. Joint-Training allows the model to maintain access to all previously encountered categories, thereby mitigating the risk of catastrophic forgetting, albeit at a computational cost. In contrast, Fine-Tuning involves updating the model exclusively with new

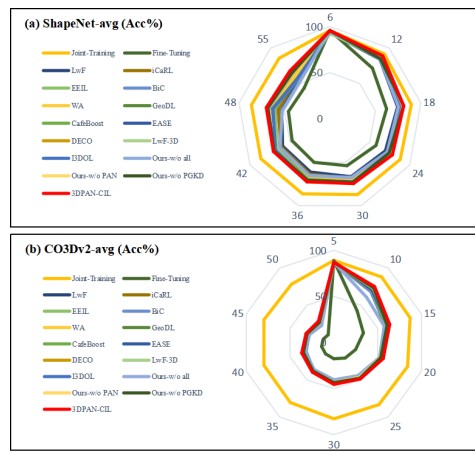 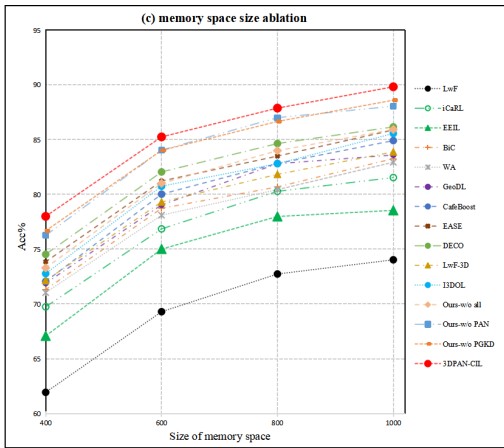

Figure 2: Method comparison and memory space size ablation on different datasets. In (a) and (b), there are 9 and 10 incremental phases respectively, and each angle of radar chart represents the number of visible categories in incremental phases. Because the results between our model and other methods are relatively low and the change trend of these methods is similar when the size of memory space is smaller than 400, we do not display them in (c).

class data, without retaining any prior samples, which can result in significant catastrophic forgetting due to the lack of rehearsal mechanisms. These benchmarks serve to delineate the upper and lower limits of incremental learning performance, respectively.

**4.3 Result comparison.** We assess our model utilizing the standard class-incremental protocol (Rebuffi et al., 2017) across above datasets. The comparative ACC results are presented in Table 1, Figure 2(a)-(b), supplementary Figure S2(a)-(b), Tables S3, S4, S5 and S6, with each task comprising a balanced subset of classes from the respective dataset. Notably, in the context of average partitioning (ModelNet40-avg, ShapeNet-avg, ScanObjectNN-avg, ScanNet-avg and CO3Dv2-avg), our model demonstrates the superior average accuracy compared to state-of-the-art methods, achieving improvements of 4.5%, 3.47%, 2.6%, 1.25% and 0.55%, respectively. Furthermore, the average forgetting rate (AFR) is reduced by 1.47%, 0.89%, 0.25%, 0.58% and 0.04%, respectively. In the case of synthetic datasets, our proposed 3DPAN-CIL exhibits significant enhancements. Particularly on the CO3Dv2 dataset, challenges such as noise, data incompleteness, and point cloud sparsity in real-world data hinder the ability of most existing models to effectively extract features from point clouds, thereby limiting the performance improvement. In contrast, our model excels on real datasets, attributed to the implementation of optimal transport-based feature prototype migration.

Table 1: Result comparison on ModelNet40-avg and partial ablation results (%).

| Method | Number of visible categories in incremental phases | | | | | | | | | | Acc$_{avg}$ | AFR |
|---|---|---|---|---|---|---|---|---|---|---|---|---|
| | 4 | 8 | 12 | 16 | 20 | 24 | 28 | 32 | 36 | 40 | | |
| Joint-Training | 99.51 | 97.46 | 96.53 | 95.95 | 93.51 | 92.44 | 91.30 | 90.10 | 89.99 | 88.53 | 93.53 | 0.69 |
| Fine-Tuning | 99.50 | 54.07 | 42.00 | 33.06 | 26.62 | 21.15 | 17.96 | 15.79 | 14.32 | 12.97 | 33.74 | 9.77 |
| LwF | 97.78 | 90.12 | 83.53 | 72.43 | 68.55 | 63.89 | 59.38 | 57.52 | 52.45 | 47.26 | 69.29 | 8.12 |
| iCaRL | 98.77 | 94.70 | 89.24 | 82.32 | 76.95 | 71.93 | 69.35 | 65.14 | 62.51 | 57.62 | 76.85 | 7.28 |
| EEIL | 97.60 | 93.81 | 87.53 | 81.67 | 78.21 | 74.72 | 69.22 | 62.42 | 56.81 | 48.12 | 75.01 | 7.15 |
| BiC | 99.26 | 94.38 | 89.16 | 81.75 | 77.63 | 74.05 | 70.48 | 68.54 | 67.53 | 64.49 | 78.73 | 6.98 |
| WA | 99.01 | 94.57 | 88.62 | 79.56 | 77.39 | 73.81 | 70.20 | 67.80 | 66.49 | 63.55 | 78.10 | 7.15 |
| GeoDL | 99.13 | 93.16 | 88.89 | 81.68 | 79.96 | 75.33 | 72.56 | 68.96 | 67.59 | 63.09 | 79.04 | 6.41 |
| CafeBoost | 99.37 | 95.12 | 90.86 | 83.02 | 79.32 | 77.08 | 73.98 | 70.16 | 67.90 | 63.43 | 80.02 | 6.58 |
| EASE | 98.48 | 94.75 | 92.30 | 86.23 | 84.21 | 79.54 | 74.39 | 71.46 | 66.42 | 64.45 | 81.22 | 6.93 |
| DECO | 99.31 | 95.89 | 93.56 | 86.03 | 84.28 | 80.69 | 75.33 | 72.81 | 65.87 | 66.74 | 82.05 | 6.38 |
| LwF-3D | 98.42 | 92.35 | 89.37 | 81.41 | 79.79 | 76.26 | 73.17 | 70.02 | 68.01 | 64.23 | 79.30 | 6.53 |
| I3DOL | 99.26 | 95.49 | 90.71 | 83.03 | 81.72 | 77.34 | 74.05 | 71.07 | 68.88 | 65.95 | 80.75 | 5.49 |
| Ours-w/o all | 99.25 | 96.44 | 94.19 | 87.91 | 85.65 | 81.49 | 75.63 | 73.44 | 70.33 | 65.80 | 81.01 | 4.98 |
| Ours-w/o PAN | 99.26 | 96.83 | 94.36 | 89.47 | 86.23 | 82.02 | 76.38 | 75.24 | 73.22 | 67.42 | 84.04 | 4.79 |
| Ours-w/o PGKD | 99.26 | 96.95 | 94.40 | 88.01 | 86.77 | 82.36 | 76.51 | 75.15 | 73.45 | 67.34 | 84.02 | 4.45 |
| 3DPAN-CIL | **99.26** | **97.63** | **94.80** | **91.79** | **89.68** | **83.75** | **77.68** | **75.95** | **74.01** | **67.96** | **85.25** | **4.02** |

Note that there is a Co-Transport method (Zhou et al., 2021) which is also based on OT, and there are some recently published methods including CIL with shape pre-training (Qi et al., 2025) and model with texture amplification and CLIP (Xiang et al., 2025). For the Co-Transport method, it uses prospective and retrospective transports to determine the semantic relationship between categories, and then based on this, constructs mappings between new and old categories to guide the synthesis of classifiers for 2D vision. Our method firstly constructs a 3D prototype space through geometric feature enhancement for 3D point clouds, and then uses OT to transfer and adjust the position of new categories in this prototype space. The implementation process and application scope are fundamentally different. We provide the result comparison for above three methods and find our model is superior to these methods (because of our enhanced 3D feature extraction, batch-wise OT and prior guided knowledge distillation). For the details, please see the supplementary Table S2.

### 4.4 Ablation experiment.

**4.4.1 Module ablation.** To assess the efficacy of the prototype assisted network (PAN) module and the prior guidance knowledge distortion (PGKD) module in our 3DPAN-CIL model, we conduct a series of experiments utilizing above five datasets for module ablation analysis. The results are presented in Table 1 and supplementary Tables S3, S4, S5 and S6. Note that Ours-w/o all, Ours-w/o PAN and Ours-w/o PGKD correspond to the model's performance when PAN and PGKD modules are excluded, the PAN module is omitted, and the PGKD module is omitted, respectively. The findings indicate that the absence of PAN during the incremental learning process hinders the model's ability to receive appropriate guidance during backpropagation, resulting in misalignment due to shifts in the category prototype space. We observe an average performance decline of 1.21%, 1.67%, 1.96%, 0.84%, 1.79% for ACC and 0.77%, 1.72%, 1.2%, 1.94%, 0.31% for AFR. Furthermore, the exclusion of PGKD leads to a lack of prior knowledge regarding the ratio of new and old samples, causing the model to favor new categories. This results in an average performance reduction of 1.23%, 1.97%, 1.29%, 2.33%, 1.63% for ACC and 0.43%, 1.02%, 2.01%, 1.51%, 0.79% for AFR. In comparison to the Ours-w/o all configuration, our 3DPAN-CIL demonstrates a significant ACC enhancement ranging from 4.24% to 8.16% across these experiments, alongside an average improvement from 0.96% to 5.43% in AFR.

**4.4.2 Memory space size ablation.** To examine the influence of memory space size on our 3DPAN-CIL framework, we vary the memory space size parameter $M$ and assess the effect of the quantity of retained historical samples on the model accuracy. We conduct experiments within the ModelNet-avg scenario, modifying the original memory space size to 400, 600, 800 and 1000 features, respectively. As illustrated in Figure 2(c), an increase in memory space size correlates with a gradual enhancement in the model performance. Furthermore, the 3DPAN-CIL approach consistently outperforms other methods across the experiments conducted with memory sizes of 400, 600, 800 and 1000, thereby supporting the assertion that 3DPAN-CIL effectively mitigates the issue of catastrophic forgetting in our model when confronted with varying memory space sizes.

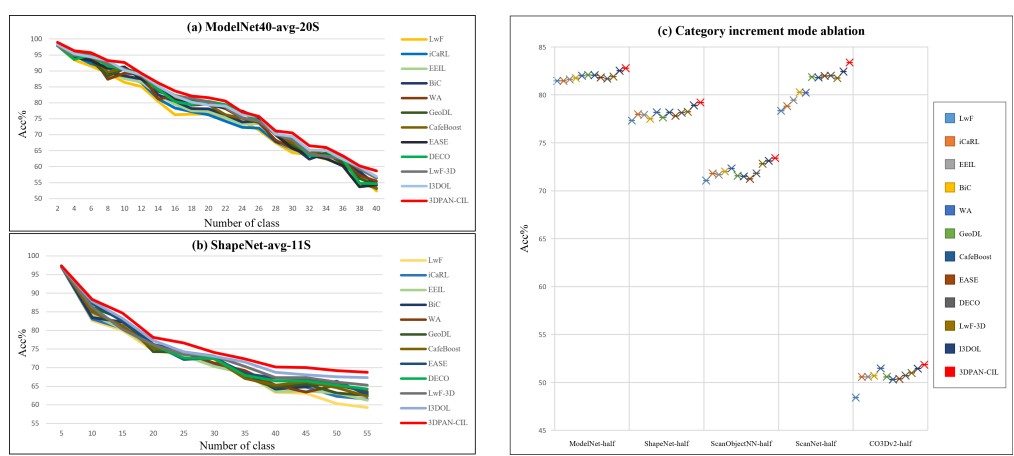

Figure 3: Incremental stage number ablation and category increment mode ablation.

**4.4.3 Incremental stage number ablation.** It is conducted to assess the efficacy of 3DPAN-CIL across varying numbers of incremental tasks, with the results illustrated in Figure 3(a)-(b). Specifically, Figure 3(a) depicts the experimental setup for ModelNet40-avg-20S, wherein the dataset is partitioned into 20 tasks, with the introduction of 2 new categories at each stage. Figure 3(b)

presents the ShapeNet-avg-11S scenario, which divides the dataset into 11 tasks, incorporating 5 new categories at each stage. The experiments on other three datasets are provided in Appendix D.4. The performance curves depicted in these figures indicate that the 3DPAN-CIL method consistently outperforms alternative approaches across various incremental stages. Furthermore, the performance metrics in the final stage remain optimal, thereby underscoring the robust adaptability of our model.

**4.4.4 Category increment mode ablation.** It is observed that a significant proportion of categories, specifically more than half, remain in the foundational stage of continuous learning. In response to this, we evaluate the performance of our 3DPAN-CIL under the over half principle in the class-incremental learning. The detailed analysis is delineated in Figure 3(c) and Appendix D.5. We also find our approach outperforms existing methods on the accuracy and AFR indices.

## 5 CONCLUSION

We introduce a novel approach (3DPAN-CIL) for class-incremental learning of 3D point clouds. The core of this work is to employ a category migration strategy based on optimal transport in the latent prototype space to effectively address the issue of catastrophic forgetting. Furthermore, we implement a knowledge distillation strategy informed by prior knowledge to counteract the classification bias that may arise from the imbalance between new and old data. In addition, we provide several geometric feature extraction and enhancement strategies for class-incremental learning of 3D point clouds. For example, we use Mini-Point for the feature extraction, and then apply Point-BERT for the feature enhancement to maintain the feature representation stability. Meanwhile, we construct the Transformer module based on masks and attention mechanism, as well as combining different maximum and average poolings, to effectively handle noise and occlusion issues in point cloud data. Additionally, we use the Sinkhorn distance calculation in OT to keep the inter-category geometric relationship in the feature prototype space. These steps enable the proposed method to remain stable geometric feature relationship during the incremental learning process of 3D point clouds. Our method has rigorously been evaluated across multiple datasets, demonstrating superior performance compared to existing state-of-the-art methods. Furthermore, our model has a generality advantage which can be extended and applied in other 3D tasks (such as segmentation and registration), in addition to 2D image domain.

The current study acknowledges certain limitations that could be addressed in future endeavors. Although our model demonstrates improvements over other advanced methods when applied to the real point cloud dataset CO3Dv2, it still encounters issues related to catastrophic forgetting, resulting in a decline in performance, particularly when dealing with sparse and incomplete point clouds for the incremental learning task associated with small sample classes. We downsample models at different ratios (such as decreasing by 20% each time) on ModelNet dataset, and find the CIL performance also continues to decline. Especially, when the model's missing degree decreases to 40-50%, the CIL performance declines significantly. For other datasets with large-scale class scenarios, we can firstly apply the downsampling technique to reduce the number of point clouds in each model, and then use our model for incremental classification learning. Additionally, for newly introduced 3D point clouds, it is worth combining with the pre-training method or vision-language model such as CLIP to solve the few-shot class-incremental learning. Finally, although we have successfully provided the proof of Sinkhorn distance in the continuous form for the first time, how to introduce more mathematical analyses (such as metric geometry and convex analysis theory) for theoretical understanding of OT and promote a deep and wide exploration of OT application will be one of our future work.

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

# APPENDIX

## A PROOF

When two distributions are represented in a continuous form, the Sinkhorn distance still satisfies the distance axiom condition for the approximate optimal solution of the OT problem.

For two distributions denoted as $\mathbf{u}_p \in \mathbb{R}^k$ and $\mathbf{v}_p \in \mathbb{R}^k$, the difference between them is articulated through the Sinkhorn distance associated with $\mathbf{u}_p$ and $\mathbf{v}_p$, as follows

$$W_{\epsilon,p}^p(\mathbf{u}_p, \mathbf{v}_p) = \inf_{\pi \in \Pi_\epsilon(\mathbf{u}_p, \mathbf{v}_p)} \int_{\mathbb{R}^k \times \mathbb{R}^k} d(\mathbf{u}_p, \mathbf{v}_p) \, d\pi, \tag{A.1}$$

where $\pi$ denotes the joint distribution function corresponding to $\mathbf{u}_p$ and $\mathbf{v}_p$, $\Pi(\mathbf{u}_p, \mathbf{v}_p)$ represents the set of all joint distributions whose marginal distributions are $\mathbf{u}_p$ and $\mathbf{v}_p$, respectively, and satisfies the KL divergence constraint Kim et al. (2021) as follows

$$\Pi_\epsilon(\mathbf{u}_p, \mathbf{v}_p) = \{\pi \mid \mathrm{KL}(\pi \| \mathbf{u}_p \otimes \mathbf{v}_p) \leq \epsilon, \pi \in \Pi(\mathbf{u}_p, \mathbf{v}_p)\}, \tag{A.2}$$

$$\mathrm{KL}(\pi \| \mathbf{u}_p \otimes \mathbf{v}_p) = \iint \pi \ln\left(\frac{\pi}{\mathbf{u}_p \mathbf{v}_p}\right) dx dy. \tag{A.3}$$

$d(\mathbf{u}_p, \mathbf{v}_p)$ represents the transport cost function from point $\mathbf{u}_p$ to $\mathbf{v}_p$, formulated as the Euclidean distance.

We firstly denote $H(\pi)$ as the entropy of the joint distribution $\pi$, which is expressed as

$$H(\pi) = -\iint_{\mathbb{R}^k \times \mathbb{R}^k} \pi \log \pi \, dx dy. \tag{A.4}$$

It is easy to know that

$$\mathrm{KL}(\pi \| \mathbf{u}_p \otimes \mathbf{v}_p) = -H(\pi) + H(\mathbf{u}_p) + H(\mathbf{v}_p). \tag{A.5}$$

**Lemma 1.** Let $\epsilon > 0$, and let $\mathbf{u}$, $\boldsymbol{\lambda}$, and $\mathbf{v}$ be three probability distributions. Let $\pi_1 \in \Pi_\epsilon(\mathbf{u}, \boldsymbol{\lambda})$ and $\pi_2 \in \Pi_\epsilon(\boldsymbol{\lambda}, \mathbf{v})$ be the optimal transport plans from $\mathbf{u}$ to $\boldsymbol{\lambda}$ and from $\boldsymbol{\lambda}$ to $\mathbf{v}$, respectively. Define

$$\pi(x, z) = \int \frac{\pi_1(x, y)\pi_2(y, z)}{\lambda(y)} \, dy, \tag{A.6}$$

then $\pi(x, z) \in \Pi_\epsilon(\mathbf{u}, \mathbf{v})$.

**Proof.** First, we verify that the marginal distribution conditions are satisfied:

$$\int \pi(x, z)dz = \int \frac{\pi_1(x, y)}{\lambda(y)} \int \pi_2(y, z)dz \, dy = \int \pi_1(x, y)dy = u(x), \tag{A.7}$$

$$\int \pi(x, z)dx = \int \frac{\pi_2(y, z)}{\lambda(y)} \int \pi_1(x, y)dx \, dy = \int \pi_2(y, z)dy = \nu(z). \tag{A.8}$$

Hence, $\pi \in \Pi(u, \nu)$.

Next, we prove that

$$H(\pi) \geq H(u) + H(\nu) - \epsilon. \tag{A.9}$$

Since $\pi_1 \in \Pi_\epsilon(u, \lambda)$, we have

$$I(u, \lambda) = H(u) + H(\lambda) - H(\pi_1) \leq \epsilon. \tag{A.10}$$

For three random variables $x$, $y$ and $z$, since $x \rightarrow y \rightarrow z$ forms a Markov chain, the Data Processing Inequality (DPI) holds, and

$$H(u) + H(\nu) - H(\pi) \leq I(u, \lambda) \leq \epsilon. \tag{A.11}$$

Therefore, $\pi \in \Pi_\epsilon(u, \nu)$.

**Theorem 1.** $1_{(u \neq \nu)} \cdot W_{\epsilon,p}^p(u, \nu)$ is a distance satisfying three axiom conditions, where

$$1_{(u \neq \nu)} = \begin{cases} 1, & u \neq \nu, \\ 0, & u = \nu. \end{cases} \tag{A.12}$$

**Proof.** By the definition of $W_{\epsilon,p}^p(u, \nu)$, we have $W_{\epsilon,p}^p(u, \nu) \geq 0$. For the transport plan $\pi$ constructed in Equation (A.6) of Lemma 1, we have

$$W_{\epsilon,p}^p(u, \nu) \leq \iint d(x, z) \, d\pi \leq \iiint (d(x, y) + d(y, z)) \frac{\pi_1(x, y)\pi_2(y, z)}{\lambda(y)} \, dy \, dx \, dz$$

$$\leq \iint d(x, y) \, d\pi_1(x, y) + \iint d(y, z) \, d\pi_2(y, z) = W_{\epsilon,p}^p(u, \lambda) + W_{\epsilon,p}^p(\lambda, \nu). \tag{A.13}$$

Since the cost function $d(x, y)$ and the coupling $\pi(x, y)$ are symmetric, it follows that $W_{\epsilon,p}^p(u, \nu)$ is symmetric. At last, it is easy to know that

$$1_{(u \neq \nu)} \cdot W_{\epsilon,p}^p(u, \nu) = 0 \quad \text{if and only if } u = \nu. \tag{A.14}$$

# B  PROCEDURE OF 3DPAN-CIL

We provide the main procedure of our 3DPAN-CIL model in the following Algorithm 1.

---

**Algorithm 1** Procedure of our 3DPAN-CIL model.

---

**Input:** Task number $T$, training data $\{D_0, \cdots, D_{T-1}\}$, empty memory $M$
**Output:** Feature extractor $f_t$, Classifier $C_t$
  1: **if** $t = 0$ **then**
  2:     Initialize parameters of feature extractor $f_0$ and classifier $C_0$
  3:     Train $f_0$, $C_0$ on $D_0$ by minimizing $L_{CE}$
  4:     **if** memory $M$ is used **then**
  5:         Update memory $M$ by $D_0$
  6:     **end if**
  7: **end if**
  8: **for** $t$ in $\{1, \cdots, T-1\}$ **do**
  9:     Update classifier $C_t$
 10:     Calculate prior knowledge $\sigma$ by Section. 3
 11:     Calculate old prototypes by Eq.equation 4
 12:     Calculate $L_{k-3DCPOT}$ by Eq.equation 7
 13:     Calculate $L_{PGKD}$ by Eq.equation 9
 14:     Train $f_t$, $C_t$ on $D_t \cup M$ by Eq.equation 10
 15:     Update memory $M$ by $D_t$
 16: **end for**

---

## C    TIME COMPLEXITY ANALYSIS

The overall time complexity of our proposed model consists of three main components: the feature extraction module, the optimal transport module, and the prior guided knowledge distillation module. The detailed analysis of each component is as follows:

- **Feature Extraction Module:** This module is based on the Vision Transformer (ViT) architecture. For each Transformer encoder layer, the time complexity is $\mathcal{O}(GK^2 + G^2K)$, where $G$ is the number of sub point clouds and $K$ is the feature dimension. With $L$ layers in the Transformer encoder, the overall complexity becomes

$$\mathcal{O}\left(L \cdot (GK^2 + G^2K)\right)$$

- **Optimal Transport Module:** This module involves three steps:
  1. Constructing the class prototype of the point cloud with complexity $\mathcal{O}(NDK)$, where $N$ is the number of classes and $D$ is the number of training samples per class.
  2. Developing the optimal transport strategy for the class prototype using the Sinkhorn distance, with complexity $\mathcal{O}(TN^2K^2)$, where $T$ is the number of iterations.
  3. Guiding the migration of new prototypes according to optimal transport, with complexity $\mathcal{O}(NK + MK)$, where $M$ is the number of newly added categories.

  The overall complexity of this module is

$$\mathcal{O}(NDK + TN^2K^2 + NK + MK)$$

- **Prior Guided Knowledge Distillation Module:** This module includes a classifier and knowledge distillation:
  - The classifier consists of 3 MLP layers with complexity $\mathcal{O}(KH_1 + H_1H_2 + H_2N)$, where $H_1$ and $H_2$ are the dimensions of the intermediate layers.
  - The knowledge distillation uses KL divergence with complexity $\mathcal{O}(N)$.

  The overall complexity is

$$\mathcal{O}(KH_1 + H_1H_2 + H_2N + N)$$

Therefore, the total time complexity of our **3DPAN-CIL** model is the sum of the complexities of all modules

$$\mathcal{O}\left(L \cdot (GK^2 + G^2K) + NDK + TN^2K^2 + NK + MK + KH_1 + H_1H_2 + H_2N + N\right)$$

Considering the dominant terms, the overall complexity can be simplified to

$$\mathcal{O}(TN^2K^2)$$

Although the introduction of the optimal transport computation increases the time complexity compared to other models, it remains within an acceptable range for general point cloud model training.

In addition, we provide a practical computational performance analysis including FLOPs, training time and peak memory usage between our model and some baseline methods (EASE, DECO, and I3DOL). For the details, please see the following Table S1. In summary, the comparative analysis of FLOPs reveals that there is no significant decline in computational performance between our model and these methods. Our method takes more training time than other three methods due to the batch OT alignment (but the difference is not significant). In terms of peak memory usage, our model is higher than EASE and DECO methods, but lower than the I3DOL method.

Table S1: Comparison of practical computation consumption and resource utilization (The computer is configured with GPU A6000 and the batch size is set to 512 during the training)

| Method | FLOPs/G | Training Time in a single epoch/s | Memory Usage/G |
|---|---|---|---|
| EASE | 2.04 | 41.58 | 37.93 |
| DECO | 1.97 | 39.30 | 39.11 |
| I3DOL | 2.26 | 45.92 | 41.82 |
| 3DPAN-CIL | 2.42 | 48.09 | 40.70 |

## D  SUPPLEMENTARY EXPERIMENTAL RESULTS

### D.1  HYPERPARAMETER SETTING IN LOSS FUNCTION

We perform the parameter tuning on two hyperparameters $(\alpha_1, \alpha_2)$ of $\mathcal{L}_{k\text{-3DCPOT}}$ and $\mathcal{L}_{\text{PGKD}}$ within the loss function of our model to ascertain the optimal combination of hyperparameters. The experimental range for these values is set to $[3, 7]$. The ModelNet10 dataset is utilized for the comparative analysis of the hyperparameter settings. The incremental stage number is designated as 5, and the memory capacity is set to 300. The dataset partitioning method employed is the average partitioning, with all other settings consistent with those used in the ModelNet40-avg experimental configuration, referred to as ModelNet10-avg. The experimental results are illustrated in Figure S1. Our finding indicates that in the ModelNet10-avg experiment, the optimal combination of hyperparameters for the loss function is designated as $\alpha_1 = \alpha_2 = 5$, resulting in the model achieving an average performance peak of 78.25%. Furthermore, in various other experimental contexts, we observe that the model's average performance also reaches its optimal level when this hyperparameter configuration is applied.

Table S2: Result comparison between our model and other three methods on ModelNet

| Method | $\text{Acc}_{\text{avg}}$ | AFR | Last Acc |
|---|---|---|---|
| Co-Transport | 78.17 | 9.48 | 63.75 |
| CIL with pre-training | 78.90 | 6.64 | 67.70 |
| CIL with CLIP | 79.80 | 8.70 | 65.90 |
| 3DPAN-CIL | 85.25 | 4.02 | 67.96 |

### D.2  THE DETAILED DESCRIPTION OF DATASETS USED IN THE EXPERIMENT

The experimental dataset comprises five commonly used datasets: ModelNet Wu et al. (2015), ShapeNet Chang et al. (2015), ScanObjectNN Uy et al. (2019), ScanNet Dai et al. (2017) and CO3Dv2 Reizenstein et al. (2021). Among them, ModelNet is derived from uniform sampling of 3D CAD models, encompassing 9,843 training samples and 2,468 test samples across 40 categories. In contrast, ShapeNet represents a more extensive 3D model dataset, featuring 55 categories with a total of 35,037 training samples and 5,053 validation samples. ScanObjectNN comprises approximately 15,000 point cloud samples, representing 15 common objects. ScanNet includes over 1,500 real indoor environments, categorized into 17 groups. Lastly, CO3Dv2 presents a more complex 3D

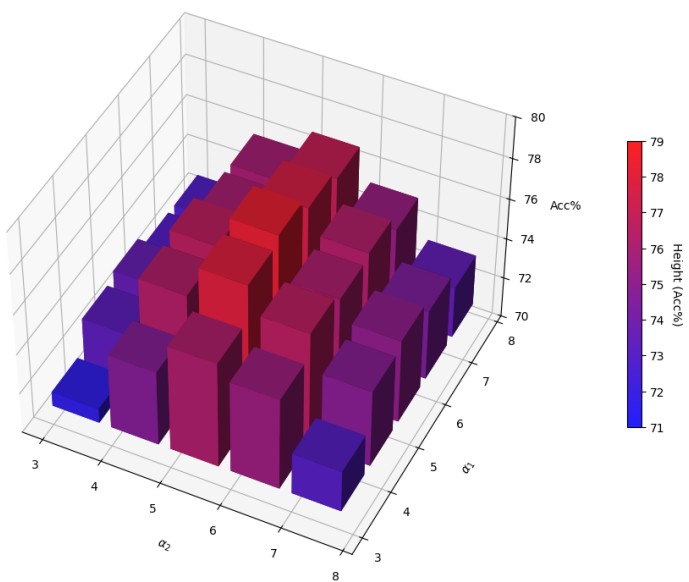

Figure S1: Comparison of hyperparameter settings for loss function.

dataset derived from real scans, encompassing 50 categories, with around 30,000 training samples and 5,000 testing samples.

## D.3 THE DETAILED DESCRIPTION OF DATASET PARTITIONING WITH AVERAGING PRINCIPLE

For five datasets, ModelNet40-avg organizes 40 categories into 10 incremental stages, introducing 4 new categories at each stage. In a similar vein, ShapeNet-avg categorizes 55 classes into 9 incremental steps, with 6 new categories added at each stage, culminating in the addition of 7 new categories in the final incremental stage. ScanObjectNN-avg divides 15 categories into an average of 5 incremental stages, incorporating 3 new categories at each stage. Additionally, ScanNet-avg segments 17 categories into 6 incremental stages, with 3 new categories introduced at each stage and 2 new categories added in the final stage. Lastly, CO3Dv2-avg partitions 50 categories into an average of 10 incremental stages, with 5 new categories added at each stage.

## D.4 INCREMENTAL STAGE NUMBER ABLATION FOR OTHER THREE DATASETS

Figure S3(a) illustrates the ScanObjectNN-avg-7S experiment, characterized by the division of the dataset into 7 tasks, with 2 new categories added at each stage and an additional 3 new categories introduced in the final task stage. Figure S3(b) outlines the ScanNet-avg-8S experiment, which similarly divides the dataset into 8 tasks, adding 2 new categories at each stage and 3 new categories in the final task stage. Lastly, Figure S3(c) details the CO3Dv2-avg-7S experiment, where the dataset is segmented into 7 tasks, with 7 new categories added at each stage and 8 new categories in the final task stage.

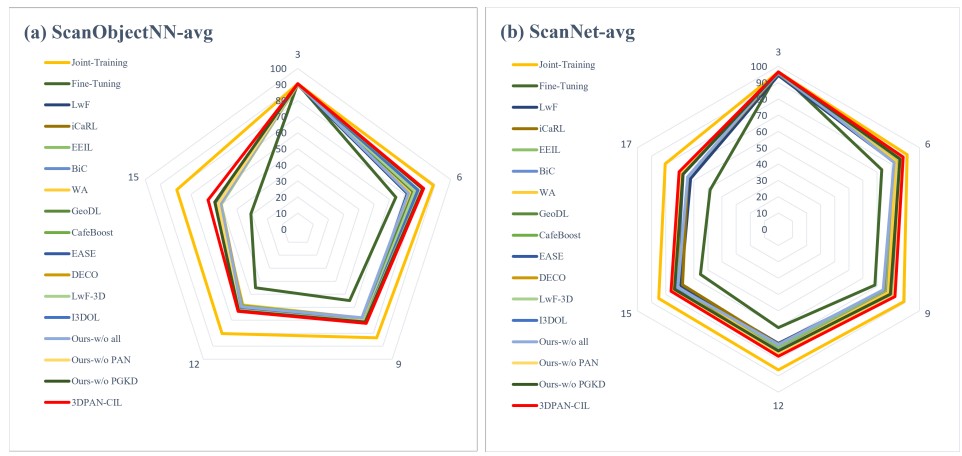

Figure S2: Different method comparison on ScanNet and ScanObjectNN datasets.

Table S3: Comparison on ShapeNet-avg and partial ablation results (%).

| Method | Number of visible categories in incremental phases | | | | | | | | | Acc$_{avg}$ | AFR |
|---|---|---|---|---|---|---|---|---|---|---|---|
| | 6 | 12 | 18 | 24 | 30 | 36 | 42 | 48 | 55 | | |
| Joint-Training | 95.60 | 91.63 | 89.58 | 87.99 | 87.28 | 86.51 | 86.43 | 86.37 | 85.62 | 88.56 | 0.48 |
| Fine-Tuning | 95.60 | 71.62 | 62.37 | 57.66 | 54.03 | 50.20 | 47.31 | 45.64 | 43.44 | 58.65 | 9.81 |
| LwF | 93.28 | 84.24 | 74.46 | 69.41 | 66.75 | 61.41 | 59.21 | 53.06 | 53.27 | 68.34 | 4.15 |
| iCaRL | 93.10 | 86.89 | 77.62 | 72.05 | 68.64 | 63.87 | 60.96 | 56.99 | 55.56 | 70.63 | 4.87 |
| EEIL | 93.59 | 87.24 | 77.57 | 73.62 | 69.98 | 66.72 | 65.39 | 60.51 | 55.82 | 72.27 | 5.29 |
| BiC | 93.83 | 87.11 | 77.75 | 74.07 | 70.66 | 68.73 | 65.73 | 63.91 | 60.31 | 73.57 | 5.72 |
| WA | 93.96 | 87.07 | 77.12 | 74.90 | 71.30 | 68.43 | 65.69 | 63.46 | 59.76 | 73.52 | 4.67 |
| GeoDL | 92.77 | 87.22 | 79.20 | 77.03 | 74.30 | 71.03 | 65.61 | 63.18 | 56.05 | 74.04 | 3.68 |
| CafeBoost | 93.24 | 88.40 | 80.68 | 76.28 | 72.72 | 69.89 | 65.85 | 61.91 | 53.54 | 73.61 | 3.84 |
| EASE | 93.64 | 88.32 | 78.32 | 76.38 | 72.36 | 72.13 | 66.67 | 62.37 | 53.29 | 73.72 | 3.25 |
| DECO | 93.26 | 87.16 | 79.54 | 76.21 | 72.64 | **72.76** | 66.42 | 62.18 | 55.42 | 73.95 | 3.16 |
| LwF-3D | 93.12 | 85.18 | 79.23 | 75.32 | 72.38 | 72.35 | 65.98 | 62.37 | 55.35 | 73.48 | 3.42 |
| I3DOL | 94.71 | 86.79 | 78.84 | 76.86 | 73.28 | 71.53 | 67.98 | 62.69 | 55.63 | 74.26 | 2.03 |
| Ours-w/o all | 95.60 | 84.47 | 75.30 | 71.73 | 67.57 | 65.29 | 62.41 | 52.38 | 51.34 | 69.57 | 6.57 |
| Ours-w/o PAN | 95.60 | 86.15 | 80.38 | 76.48 | 72.80 | 71.20 | 69.67 | 68.64 | 63.60 | 76.06 | 2.86 |
| Ours-w/o PGKD | 95.60 | 85.07 | 81.08 | 74.09 | 72.85 | 70.86 | 69.71 | 68.17 | 64.38 | 75.76 | 2.16 |
| 3DPAN-CIL | **95.60** | **89.02** | **81.21** | **78.15** | **74.63** | 72.59 | **70.77** | **69.99** | **67.60** | **77.73** | **1.14** |

Table S4: Comparison on ScanObjectNN-avg and partial ablation results (%).

| Method | Number of visible categories in incremental phases | | | | | Acc$_{avg}$ | AFR |
|---|---|---|---|---|---|---|---|
| | 3 | 6 | 9 | 12 | 15 | | |
| Joint-Training | 90.43 | 88.69 | 83.42 | 80.23 | 79.23 | 84.40 | 0.42 |
| Fine-Tuning | 90.43 | 64.16 | 54.82 | 44.95 | 30.76 | 57.02 | 9.77 |
| LwF | 89.88 | 71.65 | 69.50 | 60.23 | 51.24 | 68.50 | 6.99 |
| iCaRL | 89.93 | 74.89 | 68.58 | 59.88 | 51.63 | 68.98 | 7.16 |
| EEIL | 89.92 | 76.03 | 68.23 | 59.21 | 52.12 | 69.10 | 5.38 |
| BiC | 90.02 | 75.36 | 68.40 | 60.26 | 53.25 | 69.46 | 6.37 |
| WA | 90.44 | 73.60 | 68.34 | 58.15 | 51.75 | 68.46 | 3.68 |
| GeoDL | 89.82 | 75.58 | 68.33 | 58.96 | 52.46 | 69.03 | 4.29 |
| CafeBoost | 89.99 | 77.68 | 68.60 | 58.79 | 51.77 | 69.37 | 5.81 |
| EASE | 90.32 | 77.39 | 69.42 | 59.41 | 51.21 | 69.55 | 2.99 |
| DECO | 89.86 | 77.35 | 69.76 | 59.25 | 50.11 | 69.27 | 3.15 |
| LwF-3D | 89.96 | 76.58 | 70.11 | 60.24 | 50.32 | 69.44 | 3.68 |
| I3DOL | 90.34 | 78.74 | 71.02 | 61.34 | 52.67 | 70.82 | 2.89 |
| Ours-w/o all | 90.53 | 72.34 | 67.98 | 59.69 | 50.76 | 68.26 | 6.25 |
| Ours-w/o PAN | 90.53 | 81.29 | 70.99 | 62.36 | 52.12 | 71.46 | 3.84 |
| Ours-w/o PGKD | 90.53 | 81.72 | 71.31 | 62.74 | 54.36 | 72.13 | 4.65 |
| 3DPAN-CIL | **90.53** | **82.32** | **72.36** | **63.18** | **58.69** | **73.42** | **2.64** |

Table S5: Comparison on ScanNet-avg and partial ablation results (%).

| Method | Number of visible categories in incremental phases | | | | | | Acc$_{avg}$ | AFR |
| | 3 | 6 | 9 | 12 | 15 | 17 | | |
|---|---|---|---|---|---|---|---|---|
| Joint-Training | 96.56 | 91.36 | 88.99 | 86.37 | 84.82 | 80.31 | 88.07 | 1.99 |
| Fine-Tuning | 96.56 | 73.25 | 68.39 | 60.36 | 55.31 | 48.36 | 67.04 | 8.97 |
| LwF | 94.36 | 82.33 | 75.10 | 70.11 | 69.21 | 62.36 | 75.58 | 7.26 |
| iCaRL | 95.68 | 83.65 | 75.23 | 70.47 | 68.04 | 64.33 | 76.23 | 6.47 |
| EEIL | 95.68 | 83.24 | 76.85 | 72.13 | 72.02 | 68.21 | 78.02 | 6.56 |
| BiC | 95.47 | 83.23 | 77.55 | 74.41 | 71.62 | 70.16 | 78.74 | 5.06 |
| WA | 96.12 | 83.47 | 77.86 | 74.70 | 71.51 | 69.84 | 78.92 | 6.10 |
| GeoDL | 96.06 | 86.15 | 80.39 | 76.28 | 71.73 | 70.07 | 80.11 | 5.18 |
| CafeBoost | 96.33 | 86.54 | 80.41 | 76.80 | 72.48 | 69.85 | 80.40 | 6.35 |
| EASE | 96.33 | 86.35 | 81.27 | 76.27 | 72.32 | 67.52 | 80.01 | 5.68 |
| DECO | 96.42 | 86.39 | 81.36 | 76.21 | 72.12 | 68.99 | 80.25 | 4.49 |
| LwF-3D | 96.32 | 85.66 | 80.12 | 76.01 | 72.95 | 69.54 | 80.10 | 3.61 |
| I3DOL | **96.57** | 86.58 | 82.08 | 77.05 | 73.26 | 69.27 | 80.80 | 2.76 |
| Ours-w/o all | 96.55 | 82.01 | 74.32 | 70.64 | 71.48 | 64.32 | 76.55 | 6.92 |
| Ours-w/o PAN | 96.55 | 87.92 | 82.11 | 76.27 | 75.49 | 68.92 | 81.21 | 4.12 |
| Ours-w/o PGKD | 96.55 | 86.23 | 79.53 | 74.68 | 73.66 | 67.69 | 79.72 | 3.69 |
| 3DPAN-CIL | 96.56 | **88.42** | **82.77** | **78.01** | **76.10** | **70.45** | **82.05** | **2.18** |

Table S6: Comparison on CO3Dv2-avg and partial ablation results (%).

| Method | Number of visible categories in incremental phases | | | | | | | | | | Acc$_{avg}$ | AFR |
| | 5 | 10 | 15 | 20 | 25 | 30 | 35 | 40 | 45 | 50 | | |
|---|---|---|---|---|---|---|---|---|---|---|---|---|
| Joint-Training | 89.14 | 88.01 | 87.00 | 84.13 | 83.24 | 82.73 | 80.44 | 79.84 | 79.89 | 78.25 | 83.27 | 1.84 |
| Fine-Tuning | 88.78 | 42.80 | 33.72 | 24.76 | 20.79 | 17.58 | 15.22 | 14.02 | 12.10 | 10.49 | 28.03 | 12.9 |
| LwF | 87.04 | 68.36 | 59.33 | 53.32 | 46.28 | 41.35 | 37.41 | 31.88 | 28.42 | 26.12 | 47.95 | 7.68 |
| iCaRL | 87.62 | 69.96 | 59.82 | 54.87 | 46.55 | 41.95 | 38.54 | 32.10 | 29.54 | 26.26 | 48.72 | 7.24 |
| EEIL | 87.20 | 68.36 | 60.28 | 54.35 | 46.67 | 41.47 | 38.24 | 33.28 | 29.62 | 26.43 | 48.59 | 6.68 |
| BiC | 87.65 | 68.49 | 61.54 | 54.98 | 46.85 | 42.32 | 37.41 | 31.69 | 29.41 | 25.41 | 48.58 | 6.53 |
| WA | 87.06 | 70.98 | 62.24 | 55.62 | 47.25 | 44.88 | 38.15 | 35.16 | 30.20 | 25.33 | 49.69 | 5.69 |
| GeoDL | 87.06 | 71.70 | 60.76 | 55.64 | 47.12 | 43.47 | 38.64 | 34.98 | 30.68 | 26.89 | 49.69 | 5.24 |
| CafeBoost | 87.07 | 71.21 | 60.44 | 54.99 | 46.63 | 43.69 | 38.24 | 35.64 | 31.14 | 26.82 | 49.59 | 6.84 |
| EASE | 87.68 | 70.67 | 61.97 | 55.64 | 46.41 | 43.84 | 39.12 | 35.87 | 31.33 | 27.22 | 49.98 | 7.28 |
| DECO | 87.04 | 72.36 | 62.39 | 56.02 | 47.21 | 44.85 | 39.01 | 36.47 | 30.58 | 27.34 | 50.33 | 6.18 |
| LwF-3D | 87.13 | 73.20 | 61.20 | 55.87 | 47.55 | 44.71 | 39.31 | 36.42 | 31.11 | 27.52 | 50.40 | 5.94 |
| I3DOL | **87.14** | 73.55 | 62.67 | 56.12 | 47.68 | 45.01 | 39.41 | 36.23 | 31.33 | 28.34 | 50.75 | 5.34 |
| Ours-w/o all | 87.07 | 60.61 | 57.45 | 52.18 | 44.35 | 40.39 | 36.35 | 32.49 | 27.82 | 22.82 | 46.15 | 8.42 |
| Ours-w/o PAN | 87.07 | 74.08 | 61.42 | 52.82 | 46.76 | 43.06 | 38.30 | 34.71 | 30.44 | 26.41 | 49.51 | 5.51 |
| Ours-w/o PGKD | 87.07 | 73.62 | 60.79 | 53.41 | 47.86 | 43.47 | 38.35 | 34.76 | 30.56 | 26.82 | 49.67 | 5.99 |
| 3DPAN-CIL | 87.07 | **75.09** | **63.71** | **56.28** | **48.83** | **45.40** | **39.72** | **36.71** | **31.74** | **28.48** | **51.30** | **5.20** |

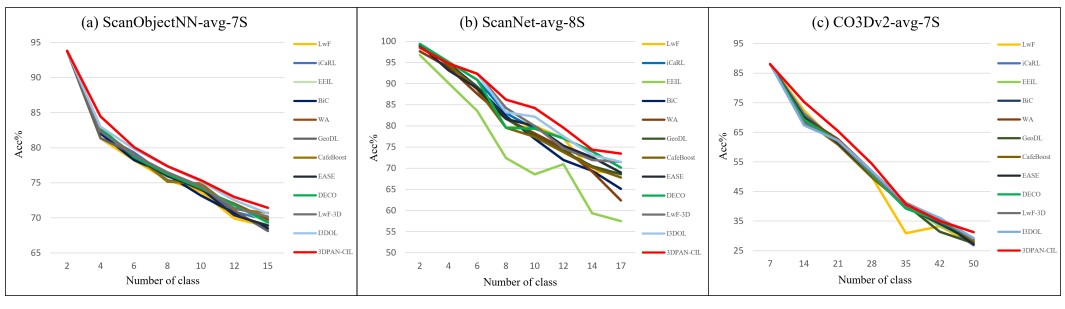

Figure S3: Ablation experiments at different incremental stages on ScanObjectNN, ScanNet, and CO3Dv2 datasets.

### D.5 CATEGORY INCREMENT MODE ABLATION UNDER THE OVER HALF INCREMENTAL PRINCIPLE

In the ModelNet40-half dataset, the foundational stage comprises 20 out of 40 categories, with the remaining 20 categories distributed evenly across 5 incremental stages, each introducing 4 new

categories, culminating in a total of 6 stages. The memory capacity for this configuration is set at 600. Similarly, in the ShapeNet-half dataset, the foundational stage includes 25 categories, while the remaining 30 categories are divided into 6 incremental steps, each adding 5 new categories, resulting in a total of 7 stages, also with a memory capacity of 600. In the ScanObjectNN-half dataset, the foundational stage consists of 9 categories, with the remaining 6 categories divided into 3 incremental steps, each contributing 2 new categories, leading to a total of 4 stages and a memory capacity of 100. The ScanNet-half dataset follows a similar structure, with 9 foundational categories and 8 additional categories divided into 4 incremental steps, each adding 2 new categories, resulting in 5 stages and a memory capacity of 100. Lastly, the CO3Dv2-half dataset features 25 foundational categories, with the remaining 25 categories divided into 5 incremental steps, each introducing 5 new categories, for a total of 6 stages and a memory capacity of 600. Throughout this process, all categories are randomly shuffled. The AFR comparison of category increment mode ablation for different methods is provided in Figure S4. We find our approach outperforms existing methods on the AFR index.

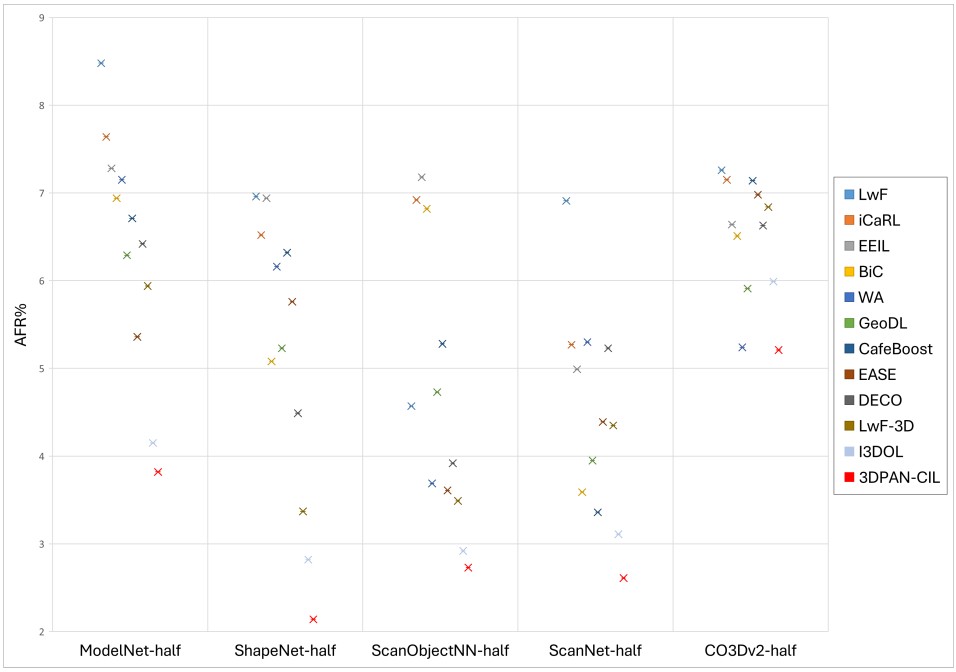

Figure S4: AFR comparison of category increment mode ablation for different methods.

