# OpenReview forum: "3DPAN-CIL: a prototype assisted network of class-incremental learning for 3D point clouds"
_ICLR.cc/2026/Conference — ICLR 2026 Conference Withdrawn Submission_

### Official Review · Reviewer_bugL · 2025-10-27

**Soundness:** 2
**Presentation:** 2
**Contribution:** 3
**Rating:** 4
**Confidence:** 4

**Summary:**

This paper investigates the application of optimal transport (OT) in incremental learning for 3D point clouds. It proposes a novel framework and provides detailed experimental validation. The study contains interesting findings and contributes valuable insights. However, there are certain aspects that the authors need to address.

**Strengths:**

1. This paper adheres to the standard academic introduction paradigm, outlining the broad context, specific problems, and limitations of existing approaches.

2. It clearly identifies the core challenges in incremental learning for 3D point clouds: catastrophic forgetting and the imbalance between new and old categories. This demonstrates that the authors have indeed recognised the critical pain points within this field.

**Weaknesses:**

1. It is recommended to elaborate more thoroughly on the fundamental differences between incremental learning for 3D point clouds and that for 2D images.
2. Whilst the paper identifies shortcomings in existing approaches, a more thorough comparative analysis is recommended. It is suggested to conduct a detailed examination of one or two representative 3D or 3D-adapted incremental learning methods (such as those based on replay or knowledge distillation), elucidating the specific challenges they encounter when processing 3D point cloud data streams. These challenges should encompass aspects such as feature representation stability, preservation of inter-category geometric relationships, and robustness against noise and occlusion.
3. The paper claims to ‘introduce OT for the first time’, which is a strong assertion requiring more thorough contextual support and differentiated analysis. How does your approach fundamentally differ from other works that may have applied OT, in terms of problem formulation, objective function design, or integration with 3D data characteristics?
4. ‘prototype’ and ‘OT alignment’ are not novel concepts in 2D vision. Therefore, the paper's argumentation regarding innovation should be strengthened.

**Questions:**

1. The paper's use of multiple datasets for evaluation is commendable. It is recommended to briefly justify the selection of these specific datasets (e.g., ModelNet, ShapeNet, ScanObjectNN) within the main text.
2. To ensure the timeliness and comprehensiveness of experimental comparisons, it is strongly advised to incorporate comparative analyses with recently published, directly relevant 3D incremental learning studies or closely related work within the relevant sections.

---

> ### Author Response · Authors · 2025-11-26
>
> 1. It is recommended to elaborate more thoroughly on the fundamental differences between incremental learning for 3D point clouds and that for 2D images.
>
> Response: Compared to 2D images, CIL for 3D point clouds meets following issues: (1) Unorderness and irregularity in 3D point clouds with noise, (2) Geometric preservation in 3D data with incompleteness and occlusion. How to preserve geometric feature and relationship is the main challenge in the CIL of 3D point clouds. We add the description in lines 101-103.
>
> 2. Whilst the paper identifies shortcomings in existing approaches, a more thorough comparative analysis is recommended. It is suggested to conduct a detailed examination of one or two representative 3D or 3D-adapted incremental learning methods (such as those based on replay or knowledge distillation), elucidating the specific challenges they encounter when processing 3D point cloud data streams. These challenges should encompass aspects such as feature representation stability, preservation of inter-category geometric relationships, and robustness against noise and occlusion.
>
> Response: We provide the comparison between our model and some representative methods (LwF based on KD and iCaRL based on replay) in the experiment. Furthermore, we add the detailed analysis how to deal with geometric challenges in our model in lines 502-509 as:
>
>    We use Mini-Point for feature extraction and apply Point-BERT for feature enhancement to maintain the feature representation stability. We construct the Transformer based module with masks and attention mechanism to effectively handle noise and occlusion issues in point clouds. In addition, we use the Sinkhorn distance calculation in OT to remain the inter-category geometric relationship in the prototype space. These steps enable our model to maintain stable geometric feature during the CIL of 3D point clouds.
>
> 3. The paper claims to ‘introduce OT for the first time’, which is a strong assertion requiring more thorough contextual support and differentiated analysis. How does your approach fundamentally differ from other works that may have applied OT, in terms of problem formulation, objective function design, or integration with 3D data characteristics?
>
> Response: We modify the related description in the revised version. For Co-Transport (ACM MM 2021), its main idea is that it uses prospective and retrospective transports to determine semantic relationship between categories, and then based on this, constructs mappings between new and old categories to guide the synthesis of classifiers for 2D vision.
>
>    Our model constructs a prototype space through geometric feature enhancement for 3D point clouds, and then uses batch-wise OT to guide the migration of new categories in the prototype space with prior guided knowledge distillation (KD). The implementation process (including problem formulation and objective function design) and application scope are different, and our result is superior to that of Co-Transport. We add the description in the last paragraph of Sect. 4.3.
>
> 4. ‘prototype’ and ‘OT alignment’ are not novel concepts in 2D vision. Therefore, the paper's argumentation regarding innovation should be strengthened.
>
> Response: The main novelty of our model is to wisely combine two concepts by using OT alignment in the prototype space. In addition, we reduce memory usage limitation by batch-wise OT and solve the data bias in training by prior guided KD, achieving excellent CIL of 3D point clouds. We add the description at the end of introduction section.
>
> Another novelty is that we provide a complete proof of Sinkhorn distance from the discrete case (Cuturi’s work, NIPS 2013) to the continuous case in a more general form (for the first time, see Appendix A), which can introduce more theoretically analysis on OT and feature preservation in KD as our future work. We add the description at the end of conclusion section.
>
> Questions:
>
> 1. The paper's use of multiple datasets for evaluation is commendable. It is recommended to briefly justify the selection of these specific datasets (e.g., ModelNet, ShapeNet, ScanObjectNN) within the main text.
>
> Response: Based on data scale, model type and application scenario, we use five datasets for testing and validation. We add the description about datasets in Sect. 4.1.
>
> 2. To ensure the timeliness and comprehensiveness of experimental comparisons, it is strongly advised to incorporate comparative analyses with recently published, directly relevant 3D incremental learning studies or closely related work within the relevant sections.
>
> Response: We add two latest papers (2025) for experimental comparison and analysis with another Co-Transport method (ACM MM 2021). Compared to OT based Co-Transport for 2D vision, our result is superior to this method. Compared to two latest methods based on pre-training and CLIP, our model also keeps superior results. We add the description in the last paragraph of Sect 4.3 and line 103.

---

### Official Review · Reviewer_VUh9 · 2025-10-30

**Soundness:** 3
**Presentation:** 2
**Contribution:** 2
**Rating:** 4
**Confidence:** 4

**Summary:**

To address the challenge of continuously arriving 3D point cloud data, this paper proposes 3DPAN-CIL, a class-incremental learning method. This method leverages prototype-based optimal transport and prior-guided knowledge distillation, which effectively mitigates catastrophic forgetting and the imbalance between old and new knowledge. And it shows great performance improvement in the experiment.

**Strengths:**

1. This article demonstrates a certain level of innovation. The authors propose a novel optimal transport method called k-3DCPOT. This approach leverages optimal transport to align the old categories prototype generated by the network at stage t with those acquired at stage t-1, thereby preserving discriminative capability for previously learned categories and mitigating catastrophic forgetting during the incremental learning process. And they also propose the Priori guided knowledge distillation which can preserve the retention of old knowledge.

2. The article demonstrates clear and logical writing. It is not difficult to understand the paper.

3. The comparative experiment is well-designed and comprehensive, featuring extensive evaluations against numerous existing works across multiple datasets.

**Weaknesses:**

1. The figure illustrations in this article still require improvement. For instance, Figure 1(a) may use $L_{k-3DCPOT}$ instead of $L_{3DCPOT}$. Additionally, more variables used in subsequent derivations, such as $Z_{t}$ and $P_{t}$, should be included in the figure to enhance readability. The visual clarity of Figures 2(a) and (b) is also insufficient, and they lack detailed explanations.

2. It lacks citations and comparison of recent work, such as [1][2], which fails to demonstrate its competitiveness.

3. The authors conducted comparative experiments focusing solely on the memory consumption of the proposed method without evaluating its computational resource consumption. Meanwhile, they noted in the appendix that the time complexity of the method scales quadratically with the number of classes, yet claimed this to be acceptable. This raises doubts about the method's performance, particularly as the authors did not test its scalability on large-scale class scenarios.

 [1] Qi, Chao, et al. "Boosting the Class-Incremental Learning in 3D Point Clouds via Zero-Collection-Cost Basic Shape Pre-Training." arXiv preprint arXiv:2504.08412 (2025).

 [2] Xiang, Tuo, et al. "Seeing 3D Through 2D Lenses: 3D Few-Shot Class-Incremental Learning via Cross-Modal Geometric Rectification." Proceedings of the IEEE/CVF International Conference on Computer Vision. 2025.

**Questions:**

1. Why choose Point-BERT as the feature extractor?
2. Have the authors considered the adaptation of this work to other 3D point cloud-related tasks?
3. Have the authors tested the method under smaller memory constraints? The minimum memory limit set in the paper was only 400 features.

---

> ### Author Response · Authors · 2025-11-26
>
> 1. The figure illustrations in this article still require improvement. For instance, Figure 1(a) may use Lk-3DCPOT instead of L3DCPOT. Additionally, more variables used in subsequent derivations, such as Zt (Zin) and Pt, should be included in the figure to enhance readability. The visual clarity of Figures 2(a) and (b) is also insufficient, and they lack detailed explanations.
>
> Response: Thanks a lot for the reviewer’s mention. We modify them in the revised version. For example, we update L3DCPOT by Lk-3DCPOT, add Zin, Zout and Pt in the figure. In the caption of Figure 2, we add more detailed explanation in (a) and (b): there are 9 and 10 incremental phases respectively, and each angle of radar chart represents the number of visible categories in incremental phases.
>
> 2. It lacks citations and comparison of recent work, such as [1][2], which fails to demonstrate its competitiveness.
>
> Response: We add the experimental comparison between our model and two methods. It is found that although R [1] uses the shape pre-training and R [2] applies texture amplification with CLIP models for incremental learning, our method still maintains superior results because of our enhanced 3D feature extraction, batch-wise OT and prior guided KD. For instance, on the ModelNet dataset, the average accuracy (85.25) of our method is higher than those of R[1, 2] (78.9 and 79.8 respectively). For the average forgetting rate, our result (4.02) is also better than those of R[1, 2] (6.64 and 8.70 respectively). For the details, please see the description in the last paragraph of Sect. 4.3 and supplementary Table S2 in the revised version.
>
> We believe that if we add pre-training and CLIP modules to our current model, the CIL ability could be further improved, which is also one of our future work. We add the relevant description in line 522.
>
> [1] Qi, Chao, et al. "Boosting the Class-Incremental Learning in 3D Point Clouds via Zero-Collection-Cost Basic Shape Pre-Training." arXiv preprint arXiv:2504.08412 (2025).
>
> [2] Xiang, Tuo, et al. "Seeing 3D Through 2D Lenses: 3D Few-Shot Class-Incremental Learning via Cross-Modal Geometric Rectification." Proceedings of the IEEE/CVF International Conference on Computer Vision. 2025.
>
> 3. The authors conducted comparative experiments focusing solely on the memory consumption of the proposed method without evaluating its computational resource consumption. Meanwhile, they noted in the appendix that the time complexity of the method scales quadratically with the number of classes, yet claimed this to be acceptable. This raises doubts about the method's performance, particularly as the authors did not test its scalability on large-scale class scenarios.
>
> Response: In our current work, we conduct experiments on two large-scale scanning datasets, ScanNet and CO3Dv2, and find that the training time is within an acceptable range. In addition, we add the computation performance comparison with FLOPs, training time and peak memory usage between our model and other baseline methods (EASE, DECO, and I3DOL), and find there is no significant decline in computational performance among them. Please see the related description in lines 816-822 and supplementary Table S1 in the revised version.
>
> For other datasets with large-scale class scenarios, we can firstly apply the downsampling technique to reduce the number of point clouds in each model, and then use our model for incremental classification learning. We add the related description in lines 519-521 in the conclusion section.
>
> Questions:
>
> 1. Why choose Point-BERT as the feature extractor?
>
> Response: Point-BERT is a point cloud processing model based on the Transformer architecture. Its core advantage lies in the combination of mask modeling, local information encoding and attention mechanism, which can significantly improve the ability of point cloud feature extraction. Therefore, we employ the feature extraction of 3D point clouds based on Point-BERT, which is helpful for the 3D point cloud’s prototype space construction in our method. We add the related description in lines 131-133, in Sect 3.2 of the revised version.
>
> 2. Have the authors considered the adaptation of this work to other 3D point cloud-related tasks?
>
> Response: Yes, our current work is focused on class-incremental learning, and we can further expand it to incremental segmentation and retrieval learning tasks of 3D point clouds, as one of our future work. We add the related description in lines 510-512, in the conclusion section of the revised version.
>
> 3. Have the authors tested the method under smaller memory constraints? The minimum memory limit set in the paper was only 400 features.
>
> Response: We conducted related tests and found the results between our model and other methods are relatively low. Moreover, the result change trend of all methods is similar, we did not display them in the figure 2(c). We add the description in the caption of Figure 2 in the revised version.

---

### Official Review · Reviewer_rZDG · 2025-10-30

**Soundness:** 2
**Presentation:** 2
**Contribution:** 2
**Rating:** 4
**Confidence:** 4

**Summary:**

This paper proposes 3DPAN-CIL, a class-incremental learning method for 3D point clouds that leverages optimal transport for prototype alignment and prior-guided knowledge distillation to mitigate catastrophic forgetting. The technique introduces batch-wise optimal transport with a KL-constrained Sinkhorn distance to migrate new category prototypes while preserving their relative spatial distributions with respect to old prototypes. Additionally, a sample-size prior guides knowledge distillation to address old/new class imbalance. The experimental results on ModelNet40 demonstrate state-of-the-art performance, with an average accuracy of 85.25% and an average forgetting rate of 4.02, outperforming 11 baseline methods.

**Strengths:**

1. The paper introduces OT-based prototype migration to the 3D point cloud domain with batch-wise formulation, providing theoretical grounding with a proof that Sinkhorn distance satisfies distance axioms in continuous form. This addresses the limitation that previous methods use sample-level distillation, which suffers from off-centre distillation issues.

2. The evaluation covers both synthetic (ModelNet40, ShapeNet) and real-world datasets (ScanObjectNN, ScanNet, CO3Dv2), comparing against 11 baseline methods, including both 3D-specific (LwF-3D, I3DOL) and adapted 2D methods (iCaRL, BiC, DECO). The consistent improvements across diverse datasets demonstrate robustness.

3. The paper identifies three key challenges in 3D point cloud incremental learning: distribution shift causing forgetting, prototype disparity between old and new categories, and data imbalance favoring new classes. Each technical component (OT-based migration, category prototype migration, PGKD) directly addresses one of these challenges.

4. Module ablation demonstrates that both PAN and PGKD contribute meaningfully, with the removal of all modules reducing ModelNet40-avg performance by 4.24%. Memory size ablation shows consistent gains across different memory budgets.

5. On ModelNet40-avg, the method achieves 85.25% average accuracy versus 82.05% for the next-best method (DECO), with an average forgetting rate of 4.02% versus 6.38%. Similar improvements are observed on other datasets

**Weaknesses:**

1. The paper claims to be the ”first use” of class prototype space construction and the ”first to introduce” optimal transport in this context. Prototype-based methods are standard in class-incremental learning, and optimal transport has been used in related 2D CIL work like Co-Transport[1]. The novelty lies in the specific application and batch-wise formulation for 3D, which should be more precisely stated.

2. Since the main method is based on Optimal Transport, the most relevant prior work, Co-Transport (ACM MM 2021)[1], is not included in the experimental comparisons. Without this, it’s difficult to assess the true incremental contribution over existing OT-based methods.

3. While the Appendix provides a theoretical time complexity analysis, it lacks a practical comparison of memory usage during training, or FLOPs, against the baseline methods


[1] Zhou, Da-Wei, Han-Jia Ye, and De-Chuan Zhan. ”Co-transport for class-incremental learning.” Proceedings of
the 29th ACM International Conference on Multimedia. 2021

**Questions:**

1. Can the authors provide a comparison against Co-Transport (ACM MM 2021)? If you implemented it for 3D, what were the results? If not, could you add a detailed discussion in the related work section on the key algorithmic differences that make your approach novel?

2. Can the authors provide a practical computational analysis, including training time and peak memory usage compared to key baselines like EASE, DECO, and I3DOL?

3. In the conclusions section, you mention limitations on sparse and incomplete point clouds. Could you quantify this?

4. At what level of sparsity or incompleteness does performance begin to degrade significantly?

---

> ### Author Response · Authors · 2025-11-26
>
> 1. The paper claims to be the ”first use” of class prototype space construction and the ”first to introduce” optimal transport in this context. Prototype-based methods are standard in class-incremental learning, and optimal transport has been used in related 2D CIL work like Co-Transport[1]. The novelty lies in the specific application and batch-wise formulation for 3D, which should be more precisely stated.
>
> Response: As the reviewer pointed out, our main novelty is that we combine prototype with OT by performing batch-wise OT operation in the prototype space for the CIL of 3D point clouds. In addition, we derive prior guided knowledge distillation to address the data bias inherent in the training model. We modify the description in lines 62-69.
>
>    Another novelty is that we provide a complete proof of Sinkhorn distance from the discrete case (Cuturi’s work, NIPS 2013) to the continuous case in a more general form (for the first time). This is important because it can introduce more mathematical analysis methods (such as metric geometry and convex analysis theory) for theoretical understanding of OT, promoting a deep and wide exploration of OT application. Please see Appendix A.
>
> 2. Since the main method is based on Optimal Transport, the most relevant prior work, Co-Transport (ACM MM 2021)[1], is not included in the experimental comparisons. Without this, it’s difficult to assess the true incremental contribution over existing OT-based methods.
>
> Response: We add the comparison between our model and Co-Transport, and find the result of our model is superior to this method. We provide the detailed analysis in lines 432-441 and give the comparison in supplementary Table S2.
>
> [1] Zhou, Da-Wei, Han-Jia Ye, and De-Chuan Zhan. ”Co-transport for class-incremental learning.” Proceedings of the 29th ACM International Conference on Multimedia. 2021
>
> 3. While the Appendix provides a theoretical time complexity analysis, it lacks a practical comparison of memory usage during training, or FLOPs, against the baseline methods.
>
> Response: We add the comparative analysis of FLOPs for different methods, and find the computational performance calculated by the model is 2.42G FLOPs. We also compute corresponding FLOPs for three baseline methods (EASE, DECO, and I3DOL), and find there is no significant decline of FLOPs between our model and these methods. For the details, please see the description in lines 816-822 and supplementary Table S1.
>
> Questions:
>
> 1. Can the authors provide a comparison against Co-Transport (ACM MM 2021)? If you implemented it for 3D, what were the results? If not, could you add a detailed discussion in the related work section on the key algorithmic differences that make your approach novel?
>
> Response: We add the comparison between our model and Co-Transport, and find the result of our model is superior to this method (because of enhanced 3D feature extraction, batch-wise OT and prior guided KD). We provide the detailed discussion in lines 432-441 and supplementary Table S2.
>
> 2. Can the authors provide a practical computational analysis, including training time and peak memory usage compared to key baselines like EASE, DECO, and I3DOL?
>
> Response: We add the computational analysis about training time and peak memory usage on our computer. Our model takes more training time than other three methods due to the OT alignment (but the difference is not significant). In terms of peak memory usage, our model is higher than EASE and DECO, but lower than I3DOL.
> We provide the discussion in lines 816-822 and supplementary Table S1.
>
> 3. In the conclusions section, you mention limitations on sparse and incomplete point clouds. Could you quantify this?
>
> Response: Yes. In the experimental section, we conduct quantitative analysis on the CO3Dv2 dataset. This dataset includes models with incomplete data. We find as the number of incremental stages increases, the performance of all methods will significantly decrease, but the results of our model are still better than other methods. Please see supplementary Table S6.
>
> In addition, we check the CIL result of our method on ModelNet dataset by downsampling models at different ratios (such as decreasing by 20% each time). The results show that as the model data decrease, the CIL performance also continues to decline. We add the related description in lines 517-518.
>
> 4. At what level of sparsity or incompleteness does performance begin to degrade significantly?
>
> Response: For the CO3Dv2 dataset with incomplete data, we find the overall performance decreases significantly when the number of visible categories in incremental phases is set in the range of 5 and 15. Please see supplementary Table S6 for details.
>
> For the ModelNet dataset, we downsample the model at different ratios (decreasing by 20% each time) and find when the model's missing degree decreases to 40-50%, the incremental learning performance significantly declines. We add the description in lines 518-519.

---

### Official Review · Reviewer_gxKa · 2025-11-01

**Soundness:** 3
**Presentation:** 2
**Contribution:** 3
**Rating:** 6
**Confidence:** 3

**Summary:**

The article presents a novel class-incremental learning method for
classification of 3D point clouds. The incremental learning aims to
ensure that prototypes of old classes do not change when new classes
are introduced. Optimal transport is used to this end. Prototypes of
new classes are placed among the prototypes of the old classes, while
maintaining their relative distances of older classes. This framework
is combined with a dual branch architecture and distillation to yield
the final class-incremental learning model. Experiments in 5 publicly
available datasets are presented. A large number of alternative models
are used in comparisons.

**Strengths:**

+ The approach in the prototype space is interesting.
+ The migration part given in Section 3.3.3 with equation 8 and what
  follows is indeed interesting. This concept is rather new and can
  lead to further developments.
+ Integration of distillation in their framework is potentially
  useful. It is well done.
+ The comparisons are made with a large number of models.
+ The results are very promising.
+ Ablation studies are performed to provide an insight to the
  contribution of different model components.

**Weaknesses:**

- Method description can improve, specifically
  + what is $k$ and $g$ in line 129?
  + $Z$ is used twice in the same paragraph but it does not seem to
    refer to the same thing, I guess. Please clarify.
  + What does "we employ the optimal transport method to assess
    $P_t^{old}$ and to ascertain the variance between $P_t^{old}$ and
    $P_{t-1}$? Do you use optimal transport to ensure they are the
    same?
  + $k$ is used to represent the prototype dimension on page 3 and on
    page 4 it is used to define marginal distributions. This is rather
    inconsistent.
  + What does "quantities of features associated with class $i$ mean
    in Equation 4? This equation seems like averaging of features
    across the samples belonging to class $i$.
  + Equation 5 is not understandable. The summations are over $k$ and
    $l$ but they do not appear in the $d(\cdot, \cdot)$.
- I do not understand the use of optimal transport in this paper.
  + As far as I can see, the model has one prototype vector per
    class. How does one construct the prototype of the old classes in
    phase $t$?
  + Is there an ambiguity in the correspondences between the
    prototypes between model from phase $t-1$ and the model at $t$?
  + During training phase $t$, the model does not have any samples
    belonging to the classes in phase $t-1$. So what does Equation 5
    exactly do? What are $x_j^i$?

**Questions:**

- Does the method use positional encoding in the feature extraction?
  If so, how is that defined?
- Your results are really good and I think the model makes
  sense. However, the model description is really not very good. Can
  you please provide a better description of the model, focusing on
  the points I raised above?

---

> ### Author Response · Authors · 2025-11-26
>
> 1. what is k and g in line 129?
>
> Response: g is the number of sub-point clouds obtained by applying the farthest point sampling (FPS) method to a single point cloud model, k represents the dimensional increase of point clouds from three-dimensional representation to a higher k-dimensional space. We add the description in lines 134-137 in the revised version.
>
> 2. Z is used twice in the same paragraph but it does not seem to refer to the same thing, I guess. Please clarify.
>
> Response: The first Z is the feature obtained from Mini-point as the input feature of feature extractor f, and the second is the feature obtained from feature extractor f. In order to better distinguish them, we modify two Z's as Zin and Zout, which represent the input and output features of f respectively. We modify the description in lines 134-139.
>
> 3. What does "we employ the optimal transport method to assess Ptold and to ascertain the variance between Ptold and Pt-1? Do you use optimal transport to ensure they are the same?
>
> Response: At the stage t, we calculate the distance between Pt(old) and P(t-1), simultaneously using OT in prototype space to narrow the difference between Pt(old) and P(t-1), to prevent the model from forgetting old categories. Due to P(t-1) being the prototype space from the previous stage and being frozen (not changing in prototype space), Pt(old) is calculated in real-time and migrate as the model is trained. Therefore, we use the OT method to guide Pt(old)'s migration, achieving the effect of preventing forgetting. We add the description in lines 189-193.
>
> 4. k is used to represent the prototype dimension on page 3 and on page 4 it is used to define marginal distributions. This is rather inconsistent.
>
> Response: In line 205, page 4, original k is replaced by m in the revised version.
>
> 5. What does "quantities of features associated with class i mean in Equation 4? This equation seems like averaging of features across the samples belonging to class i.
>
> Response: In Equ. 4, the feature prototype of class i is represented by averaging the features of samples belonging to that class, and let p(t-1)i be used to represent the category prototype of each class. We add the description in lines 244-245.
>
> 6. Equation 5 is not understandable. The summations are over k and l but they do not appear in the d(.,.).
>
> Response: Original k and l are written on the right side of Pi. In order to express it more clearly, we update it as d(p(t-1)k, p(t)l) in Equ. 5.
>
>
> I do not understand the use of optimal transport in this paper:
>
> 1. As far as I can see, the model has one prototype vector per class. How does one construct the prototype of the old classes in phase t?
>
> Response: We adopt a replay based strategy in stage t (retaining a small number of old samples for joint training). The purpose of using OT is to reduce the distance between old class prototypes at time t (changing when training) and class prototypes at time t-1 (frozen) in the prototype space, to alleviate catastrophic forgetting. We add the description in lines 224-225.
>
> 2. Is there an ambiguity in the correspondences between the prototypes between model from phase t-1 and the model at t?
>
> Response: In the stage t, we can separate new and old category prototypes by prior knowledge in advance, and perform OT calculation on old category prototypes and the t-1 stage’s category prototypes to reduce the difference. There is no ambiguity between prototypes in stages t-1 and t. We add the description in lines 225-226.
>
> 3. During training phase t, the model does not have any samples belonging to the classes in phase t-1. So what does Equation 5 exactly do? What are xji?
>
> Response: In the training stage t, we follow the replay-based class-incremental learning strategy, which always allows the model to store a small number of samples in the memory space for training, without the problem of no corresponding samples in the t-1 stage. In Equ. (5), we originally use ft(xji) to express the center of each category prototype in stage t. In order to express it more clearly, we update the expression as: d(p(t-1)k, p(t)l). We add the related description in lines 255-257, and modify the writing of Equ. (5).
>
>
> Questions:
>
> 1. Does the method use positional encoding in the feature extraction? If so, how is that defined?
>
> Response: We use positional encoding when employing Point-BERT for feature extraction of 3D point clouds. It takes location information as input and encodes it through the Transformer based framework for the feature extraction and enhancement representation. We add the description in lines 131-132.
>
> 2. Your results are really good and I think the model makes sense. However, the model description is really not very good. Can you please provide a better description of the model, focusing on the points I raised above?
>
> Response: Thanks for the reviewer’s encouragement. We improve the description in the revised version focusing on all points raised above.

---

### Note · Authors · 2026-01-28

I have read and agree with the venue's withdrawal policy on behalf of myself and my co-authors.

---

### Meta-Review · Area_Chair_F3jB · 2026-01-07

**Summary:**

Four reviewers evaluated this submission, resulting in a decision leaning towards rejection. One reviewer initially recommended borderline acceptance with lower confidence, while three others assigned borderline reject ratings. The paper proposes a class-incremental learning framework for 3D point clouds utilizing optimal transport for prototype alignment. While the reviewers acknowledged improvements in clarity and the inclusion of complexity analyses during the rebuttal, significant concerns persist regarding the novelty of the approach compared to existing optimal transport methods (e.g., Co-Transport) and the rigor of the experimental validation. Specifically, the authors limited their new baseline comparisons to the small-scale ModelNet10 dataset, and performance on the more complex, real-world CO3Dv2 dataset was marginal compared to prior art. Consequently, the AC recommends rejection.

**Reviewer Concerns:**

The authors successfully addressed concerns regarding notation clarity, method descriptions, and computational complexity analysis raised by Reviewers gxKa and VUh9. However, critical issues regarding comparative evaluation and novelty remain unresolved. Reviewers rZDG, VUh9, and bugL emphasized the need to compare against Co-Transport and recent state-of-the-art methods (2025); while the authors provided these comparisons, they were restricted to ModelNet10, which is considered insufficient to demonstrate robustness for modern 3D tasks. Furthermore, the novelty claim remains contentious (Reviewer bugL), as the differentiation from prior 2D approaches was not deeply established. Finally, the additional experiments on the real-world CO3Dv2 dataset requested by Reviewer rZDG showed performance levels similar to existing methods, failing to prove the practical superiority of the proposed approach.

**Reviewer Scores:**

Reviewer gxKa is expected to maintain his/her score of 6, as the authors largely resolved the specific questions regarding mathematical notation and method description that constituted the bulk of his/her review. Reviewer rZDG will likely retain his/her score of 4, because the requested comparison with Co-Transport was conducted only on a "toy" dataset (ModelNet10), and the results on the real-world CO3Dv2 dataset failed to demonstrate a significant performance advantage over existing baselines. Reviewer VUh9 is also expected to keep his/her score of 4; similar to Reviewer rZDG, they are likely to find the limitation of key baseline comparisons to ModelNet10 insufficient to prove the method's competitiveness against recent works. Reviewer bugL is expected to maintain a score of 4, as the rebuttal did not adequately justify the fundamental differences from the 2D CIL or the specific dataset selection.

---

### Decision · Program_Chairs · 2026-01-26

Reject